

# Optical properties of laboratory grown sea ice doped with light absorbing impurities (black carbon)

Amelia A. Marks[1], Maxim L. Lamare[1], and Martin D. King[1]

[1]Department of Earth Sciences, Royal Holloway University of London, Egham, Surrey, TW20 0EX, UK.

*Correspondence to:* Martin King (M.King@rhul.ac.uk)

**Abstract.** Sea ice radiative-transfer models are of great importance for prediction of future sea ice trends but they are limited by uncertainty in models and requirement for evaluation of modelled irradiance data against measured irradiance data. Presented here are the first results from the Royal Holloway sea ice simulator used to evaluate the output of the TUV-snow radiative-transfer model against the optical properties from the simulated sea ice. The sea ice simulator creates a realistic sea ice

environment where both optical (reflectance and light penetration depth (*e*-folding depth)) and physical (temperature, salinity, density) properties of a ∼30 cm thick sea ice can be monitored and measured. Using albedo and *e*-folding depth data measured from simulated sea ice, scattering and absorption cross-sections of the ice are derived using the TUV-snow model. Absorption cross-sections for the ice are highly wavelength dependent, suggesting the addition of a further absorbing impurity in the ice matching the absorption spectrum of algae. Scattering cross-sections were wavelength independent with values ranging from

0.012 and 0.032 $cm^2 kg^{-1}$ for different ice created in the simulator. Reflectance and light penetration depth (*e*-folding depth) of sea ice is calculated from the derived values of the scattering and absorption cross-section using the TUV-snow model within error of the experiment. The model is also shown to replicate ice optical properties for sea ice with an extra layer doped with black carbon, well within error of the experiment. Particulate black carbon at mass ratios of 75, 150 and 300 ng $g^{-1}$ in a 5 cm ice layer lowers the albedo by 97%, 90%, and 79% compared to clean ice at a wavelength of 500 nm.

## 1  Introduction

Rapid decline of sea ice in the Arctic is often seen as a bellwether for modern day climate change (e.g. IPCC (2013)). Model predictions of future sea ice extent have a large degree of uncertainty (e.g. IPCC (2013)). Accurate representation and recreation of the optical and physical properties of sea ice is essential to develop accurate models of sea ice. The Royal Holloway Sea Ice Simulator facility aims to create a realistic sea ice within a controlled environment with the ability to monitor both the physical

(temperature, salinity and density) and optical (nadir reflectivity and *e*-folding depth) properties of the sea ice. The results from which can be used to evaluate sea ice models.

The following study presents the first data from the RHUL sea ice simulator to validate the TUV-snow radiative-transfer model for sea ice. The TUV-snow model is a coupled atmosphere-snow/sea ice radiative-transfer model, described in detail by Lee-Taylor and Madronich (2002). The model has been used multiple times for investigations of radiative-transfer in snow and

sea ice (e.g. King et al. (2005), France et al. (2011), France et al. (2012), Reay et al. (2012), Marks and King (2013), Marks



and King (2014) and Lamare et al. (2016)) and has previously been experimentally validated for photochemistry in snow by Phillips and Simpson (2005) but it has not been experimentally validated for sea ice.

The accuracy of the TUV-snow model will be evaluated for the optical properties (reflectance and $e$-folding depths) of a sea ice compared to measured values of a sea ice grown in a laboratory. Secondly, the model will be evaluated for calculating nadir reflectance with light absorbing impurities in the sea ice, namely black carbon and algae. Sea ice typically contains impurities such as black carbon, sediment and algae (e.g.Perovich (1996)). Black carbon is an efficient absorber of solar radiation (e.g. Mitchell (1957); Highwood and Kinnersley (2006); Hansen and Nazarenko (2004); Jacobson (2001); Ramanathan and Carmichael (2008); Bond et al. (2013)) and its deposition onto, or incorporation into, sea ice has been shown through modelling calculations to decrease the surface reflectance of the sea ice, increasing melt rates (e.g. Grenfell et al. (2002); Jacobson (2004); Light et al. (1998); Ledley and Thompson (1986); Goldenson et al. (2012); Holland et al. (2012); Marks and King (2013, 2014)). To evaluate the TUV-snow model with black carbon, a commercial black carbon is added to a 5 cm surface layer of 30–50 cm thick sea ice created in the laboratory in mass-ratios of 0, 75, 150 and 300 ng g$^{-1}$ and the reflectance of sea ice measured. The experimental reflectivity is compared to a calculated reflectivity using the TUV-snow model, for the same black carbon mass-ratios. If the radiative-transfer modelling with TUV-snow can reproduce the laboratory grown ice with absorbing impurities it will allow the model to be used confidently for other sea ice types and absorbers.

Previous research on the effects of black carbon on sea ice optical properties have used radiative-transfer calculations and global climate model simulations. To the authors' knowledge there are no laboratory or field studies examining the effects of added black carbon on reducing sea ice reflectance. A related study by Hadley and Kirchstetter (2012) carried out successful laboratory experiments on artificial snow investigating the effects of black carbon on snow reflectance. The results from Hadley and Kirchstetter (2012) were used to validate the Snow, Ice and Aerosol radiation (SNICAR) model (Flanner et al., 2007) used in the 2013 IPCC report (IPCC, 2013). Similarly, Brandt et al. (2011) investigated the effect of black carbon on albedo of artificial snowpacks using snowmaking machines, showing a good match between measured values and albedos calculated from radiative transfer modelling. Peltoniemi et al. (2015) measured the effect on snow bi-directional reflectance owing to additions of Chimney soot, volcanic sand, and glaciogenic silt, demonstrating snow metamorphism caused by the addition of these particles and the subsequent impact on the albedo.

The study presented here are the first experiments with the Royal Holloway Sea Ice Simulator to evaluate the TUV snow model for undoped sea ice and the first attempt to compare measurements to calculations using a radiative transfer model of nadir reflectance owing to black carbon in sea ice.

There are five overall aims of the study: firstly to grow realistic artificial sea ice; secondly to characterise optical and physical properties of the sea ice; thirdly to use measured optical properties to recreate the irradiance within the sea ice using the TUV-snow radiative transfer model and compare modelled and measured values; fourthly to create a 5 cm layer of sea ice doped with black carbon with mass-ratios of 0, 75, 150 and 300 ng g$^{-1}$ and measure the reflectance; finally to compare the measured reflectance with black carbon in the surface layer of sea ice to calculations with the TUV-snow model.



Throughout the paper the term "experimental" refers to experiments with laboratory grown sea ice using the sea ice simulator described in section 2.1, with results being referred to as "measured" values. The term "modelled" refers to calculations from the TUV-snow radiative-transfer model, the results from which are referred to as "calculated" values.

## 2  Experimental method

The following sections will describe the design of the sea ice simulator (section 2.1), the characterisation of both the optical and physical properties of the simulated ice (section 2.2) and the creation of sea ice doped with black carbon particles (section 2.3).

### 2.1  Sea ice simulator design

The sea ice simulator is a large scale, UK-based, laboratory sea ice tank designed to replicate warmer polar temperatures, the
ocean and UV and visible wavelengths of solar radiation. Previous experiments with sea ice simulators have been carried out by, for example, Light et al. (2015); Buist et al. (2011); Papadimitriou et al. (2003); Haas et al. (1999); Polach et al. (2013); Hare et al. (2013); Grenfell and Perovich (1981). The set up of the simulator is shown in figure 1. The simulator is housed in a refrigerated shipping container (11.95 m length × 2.56 m high × 2.29 m width) which can be temperature controlled from –25°C to 25° C. Inside the container sea ice is formed in a 2000 L polyethylene cylindrical white plastic tank (1.32 m
high × 1.39 m diameter) placed on insulated pallets. Following the approach of Perovich and Grenfell (1981) a cylindrical design is utilised for the tank to help avoid mechanical stress at particular locations. A 1 cm insulating layer of black neoprene also surrounds the tank sides. A metal Unistrut frame surrounds the tank to further improve structural integrity. Black wooden boards, painted with mould resistant paint, are fixed around the Unistrut structure with 3 cm thick polystyrene insulation fitting between the wooden boards and the tank.





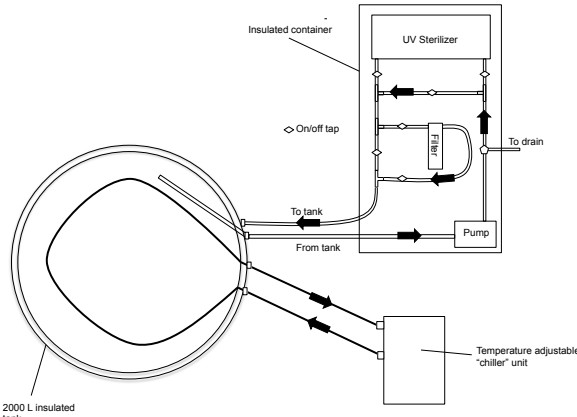

**Figure 1.** Plan of the sea ice simulator showing the 2000 L cylindrical tank in plan view and water conditioning unit in horizontal view. A closed pipe runs around the base of the tank connected to a heater unit circulating a water and glycol mixture gently warming the base of the tank. Water is circulated around the tank by a pump in an insulated container and also passed through a UV steriliser and particulate filter.

The tank is filled with a solution of Tropic Marine (Atkinson and Bingman, 1997) and water with a salinity of 32 PSU, representing Arctic ocean salinity (Boyer et al., 2013). Tropic Marine is a synthetic sea salt mixture for aquaculture containing over 70 chemical elements in typical natural concentrations representative of the ocean with the notable exception of nitrate and phosphate, to inhibit algae growth. Atkinson and Bingman (1997) show the concentrations of major cations and anions of

Tropic Marine are within 10% of seawater. Previous sea ice simulators use either sodium chloride or synthetic sea salt mixtures similar to Tropic Marine (e.g. Krembs et al. (2001), Mock et al. (2002), Papadimitriou et al. (2003) and Hare et al. (2013)).

To create circulation within the tank, ensuring temperature stratification does not occur, an Iwaki MD-10 pump circulates water at ∼10 L min$^{-1}$ at the base of the tank, as shown in figure 1. The circulated water is also pumped through a 10 $\mu$m filter to remove any particulate impurities and a UV steriliser to prevent algae growth.

Sea ice grows from surface cooling of a salt water body (Weeks, 2010). To ensure even and realistic ice growth in the tank (from the surface, downwards) a closed pipe is run around the bottom of the tank, connected to a heater unit. The heater unit contains a solution of water and pure ethylene glycol (in a 1:1 ratio) which is pumped around the pipe at a constant temperature (0°C), to warm the base of the tank and preventing freezing.

Illumination replicating the majority of shortwave solar wavelengths (350–650 nm) is provided with a set of twenty Daystar

daylight simulation fluorescent tubes and five sun-bed ultraviolet tube lights (peak illumination wavelength of ∼350 nm). Measurements of reflectance of the sea ice is a relative measurement, (i.e. the quotient of reflected radiance to incident radiance or a reflectance standard) thus the intensity-spectrum of the lamp does not have to match the solar spectrum. The lights are evenly distributed directly above the tank to provide a diffuse illumination source. The light is further diffused by opaque white boards placed around the edges of the tank. Diffuse lighting was used to simplify the measurement of $e$-folding depth and

provide a useful reflectance product.





## 2.2 Creation and characterisation of sea ice

To create sea ice an air temperature inside the container of $\sim -15°$C is maintained. Cold air is circulated within the container with fans. An additional air fan, attached to the ceiling, blows cold, ambient air at the water surface, increasing the heat flux from the ice surface, quickening ice formation and assisting the production of columnar ice (Weeks, 2010).

Sea ice is grown in the simulator for $\sim$17 days producing ice with a thickness of $\sim 30\ cm$. Temperature depth profiles and nadir reflectance of the ice were measured daily during growth (described in sections 2.2.2 and 2.2.5 respectively). Light penetration depth ($e$-folding depth) is measured at the end of the experiment as it destroys the fabric of the ice (described in section 2.2.6). The $e$-folding depth is the distance over which light intensity reduces to $\frac{1}{e}$ of its initial value.

### 2.2.1 Characterisation of sea ice physical properties

### 10   2.2.2 Temperature profiles

Temperature profiles through the sea ice are recorded daily during ice growth to give an indication of sea ice thickness and ensure that temperature stratification does not occur within the underlying seawater. The temperature is recorded via a series of thermocouples, as used by Rabus and Echelmeyer (2002); Johnston and Timco (2002); Nomurai et al. (2006). Calibrated type T thermocouples are inserted into a thin plastic sleeve and then a white polypropolyne pole at regular depths (every 2 cm)

into the water and then frozen in place during ice formation. Temperature of sea ice decreases from the surface to the base; thickness can be determined from the point where the temperature becomes constant with depth, as this can be assumed to be seawater.

### 2.2.3 Determining sea ice properties by ice coring

Cores of the ice are taken to determine sea ice properties at the conclusion of the optical experiments (section 2.2.4). The corer design was based on a CRREL report by Rand and Mellor (1985). Cores are photographed, divided into $\sim$5 cm sections and their dimensions and mass measured to derive density. Salinity is measured after melting using a Fisher Scientific seawater refractometer (cross-calibrated with an accurate ion conduction probe).

### 2.2.4 Characterisation of sea ice optical properties

### 25   2.2.5 Measuring reflectance

The nadir reflectance of the sea ice is measured daily during ice growth until the value became constant (taking between 6 and 12 days). Reflectance becomes constant once a sufficient thickness is reached that the underlying water no longer affects reflectance measurements, thus the reflectance of ice alone is measured. Upwelling radiance from the ice is measured via an

optical lens connected to a 400 $\mu$m xsr fibre optic coupled to an Ocean Optics USB 2000+ spectrometer (wavelength range:



200-850 nm, resolution: 1.5 nm FWHM, signal:noise 250:1). The optical lens is situated ~40 cm above the sea ice surface at nadir with a view footprint covering a circular area ~315 cm². The footprint is an order of magnitude larger than any surface feature on the sea ice.

To convert ice surface radiance measurements to reflectance the radiance of light from the sea ice surface measured at nadir is
ratioed to the radiance from a reference quasi-Lambertian reflector at nadir (a Spectralon panel) measured in the same location but raised 5 mm above the ice surface and under identical illumination as the sea ice.

### 2.2.6 Measuring $e$-folding depth

At the completion of the experiment the light penetration depth ($e$-folding depth) is measured. The sea ice $e$-folding depth is
measured by drilling a single hole gradually through the ice in ~5 cm increments with a sharp drill. At each depth drilled the same fibre optic is inserted into the hole and the light intensity (upwelling radiance) measured via an Ocean Optics spectrometer. The hole is a tight fit around the fibre but a thin, light diffusing disk, of white PTFE is also placed around the fibre at the ice surface to minimise any stray light entering the hole without altering the light field near the hole.

Simultaneously to the light intensity in the hole being measured ($I_{raw}$) the light intensity of another fibre optic inside a
diffusing PTFE container at the ice surface was measured ($I_{ref}$) to account for any change in the intensity of the fluorescent lights. The relative light intensity, $I_z$ , at depth, z, is then calculated using equation 1.

$$I_z = \frac{I_{raw(z)}}{I_{ref}} \tag{1}$$

The $e$-folding depth, $\epsilon$, is calculated using equation 2, where $I_z$ is the relative intensity at a depth, z, and $I_{z'}$ is intensity at the shallowest depth, $z'$. From the measured light intensity values the $e$-folding depth is calculated by fitting an exponential
curve through $I_z$ versus z data.

$$\frac{I_z}{I_{z'}} = e^{-\left(\frac{(z-z')}{\epsilon}\right)} \tag{2}$$

The $e$-folding depth measured in this work is asymptotic as the light field to the sea ice is diffuse and thus there are no near surface effects as found frequently in fieldwork (e.g. Reay et al. (2012) and references therein).

### 2.3 Creation of black carbon doped sea ice

Once the sea ice has grown to ~30 cm thick (~3 weeks of ice growth) 75 L (equivalent to a 5 cm layer) of chilled seawater doped with a known concentration of black carbon (described in section 2.3.1) is added to the surface and frozen in place forming a 5 cm black carbon bearing ice layer. Black carbon is placed within a 5 cm surface layer of the artificial ice to replicate black carbon entrainment into sea ice following melting of overlying snow as described by Grenfell et al. (2002) and Doherty et al. (2010). The new 5 cm layer of black carbon bearing seawater is left to freeze for three days and the reflectivity





of the new sea ice surface then measured daily over a week. The sea ice is then cored and density and salinity measured down the core to record the physical ice structure before and after the black carbon bearing layer is added.

At completion of the experiment the ice is melted and water is purified by filtration through a 1 $\mu$m Purtex filter to remove black carbon particulates. If any black carbon particulates were to remain the concentration would be negligible as it would be diluted by 2000 L of sea water (a dilution factor of ∼30). The whole process is repeated with other black carbon concentrations in the 5 cm layer giving a total of four mass loadings; ∼75 ng g$^{-1}$, ∼150 ng g$^{-1}$ and ∼300 ng g$^{-1}$ and a blank run with 0 ng g$^{-1}$ of black carbon. Between runs the tank is periodically bleached to remove any algae that may have grown. No algae was visible to the naked eye.

### 2.3.1 Creating atmospherically representative black carbon

To create the aqueous black carbon solutions a method from Clarke (1982) is adapted. The black carbon used, Monarch 120, is produced by Cabot Corporation to replace the discontinued Monarch 71 used by Grenfell et al. (2011). Approximately 1 g of the black carbon is added to a solution of 800 ml of pure water and 200 ml isopropanol (isopropanol aids dispersal of the black carbon in the concentrated solution) (Clarke, 1982).The solution is then placed in an ultrasonic bath for 2 hours to ensure the black carbon is fully dispersed and to break up conglomerated lumps. The solution is then suction filtered through 2 $\mu$m Nuclepore membrane filters followed by 0.8 $\mu$m Nuclepore filter to remove larger particles and ensure the final solution is representative of atmospheric black carbon i.e. particle diameter <0.8 $\mu$m (Clarke, 1982). The mass loading of black carbon in the solution is determined gravimetrically (i.e. by evaporating and weighing a proportion of the black carbon solution). Two black carbon solutions were used with mass loadings of $46 \pm 11$ $\mu$g g$^{-1}$ and $11 \pm 1.5$ $\mu$g g$^{-1}$. The uncertainties are the standard deviation of three repeated gravimetric determinations. Known amounts of solutions 1 and 2 are mixed with 75 L of artificial seawater to give overall black carbon mass-ratios detailed in table 1. The mass-ratios of black carbon are approximately 0, 75, 150 and 300 ng g$^{-1}$, these approximate values will be subsequently used in the text.

### 2.3.2 Characterisation of black carbon optical properties

The mass absorption cross-section of the black carbon used in the present study is estimated using an integrating sandwich spectrometer, described by Grenfell et al. (2011), based on Clarke (1982). The integrating sandwich spectrometer measures the absorption spectrum of particulate matter on filter samples in a diffuse radiance environment. Absorption spectra of multiple filters containing black carbon loadings are converted to a mass absorption cross-section. Mass absorption cross-sections of 0.58 and 2.1 m$^2$g$^{-1}$, ($\lambda = 610$ nm) are estimated for the black carbon placed in the artificial sea ice. The values are a factor of 3–11 smaller than the black carbon mass absorption cross-section of 6.57 m$^2$g$^{-1}$, for a wavelength of 610 nm, (Flanner et al., 2007) typically used in radiative-transfer calculations, but are similar to values used by Dang et al. (2015).





**Table 1.** Optical and physical properties of sea ice for each run. The uncertainty in sea ice density is 1 standard deviation of the average of measurements taken from the core profile. Uncertainty is not provided for the density of the top layer as this is the average of only two measurements, although the uncertainty is likely to be similar to that of the lower layer. The mass ratio of black carbon added to the surface layer is also shown. The uncertainty in the black carbon mass ratio is the uncertainty in the gravimetric method used for determining the mass ratio, as described on section 2.3.1.

| Run number | Black carbon mass-ratio added /ng g$^{-1}$ | Density of bottom (undoped) layer/ g cm$^3$ | Density of top (doped) layer/ g cm$^3$ | $\sigma_{scatt}$ bottom layer /cm$^2$ kg$^{-1}$ | $\sigma_{scatt}$ top BC layer /cm$^2$ kg$^{-1}$ |
|---|---|---|---|---|---|
| 1 | 0 | 0.91±0.084 | 0.92 | 0.315±0.040 | 0.05 |
| 2 | 77±18 | 0.91±0.059 | 0.91 | 0.235±0.041 | 0.05 |
| 3 | 153±37 | 0.92±0.044 | 1.00 | 0.115±0.004 | 0.35 |
| 4 | 305±62 | 0.95±0.050 | 0.93 | 0.126±0.016 | 0.2 |

Six known aliquots of the filtered black carbon solution described in section 2.3.1 were filtered through 0.4$\mu$m Nuclepore filters, providing filter loadings of 10.076, 15.115, 20.153, 25.191, 50.383 and 100.765 $\mu$g cm$^{-2}$. The absorbance spectra of the filters (figure 2a) is calculated using equation 3:

$$A(\lambda) = -ln\frac{I(\lambda)}{I_0(\lambda)}$$

(3)

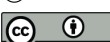



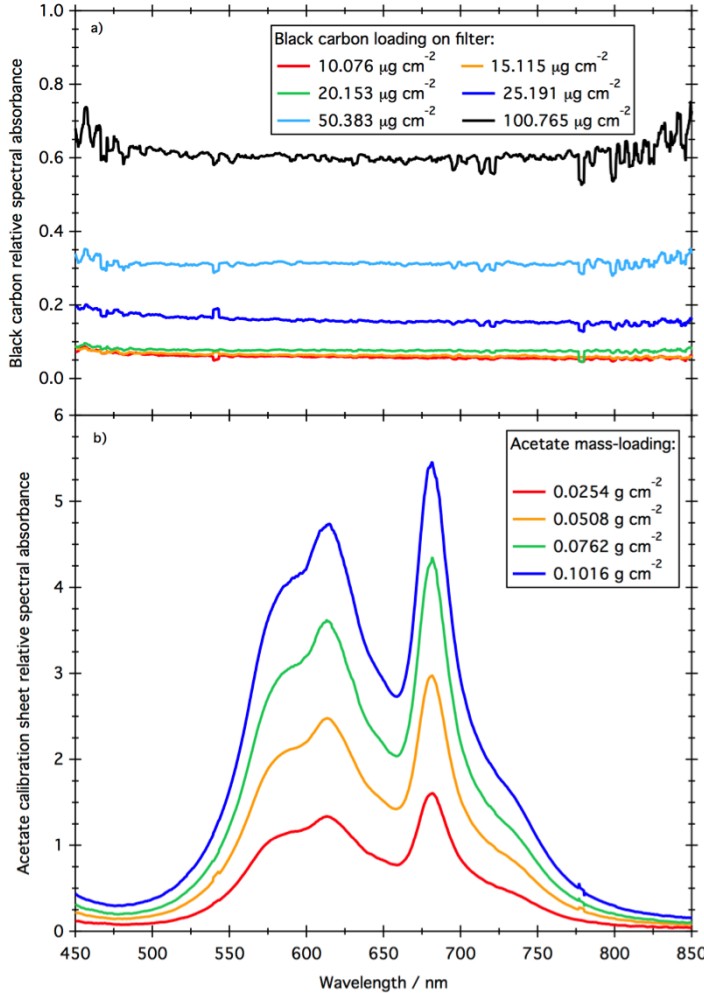

**Figure 2.** Black carbon relative absorbance. b) Acetate calibration sheet relative absorbance.

Where I is the intensity measured with the loaded filter in the integrating sandwich spectrometer, and $I_0$ is the intensity measured when a blank $0.4\mu m$ Nuclepore filter, which is measured following the same procedure as the loaded filter. To calibrate the integrating sandwich spectrometer, two sets of translucent standard plastic sheets (Light Blue Acetate film, $150\mu m$ and Light Blue Polypropylene, $100\mu m$) with measurable mass absorption coefficients are used. The sheets are placed on a 0.4 $\mu m$ Nuclepore filter and measured in the integrating sandwich spectrometer using the same method as the black carbon filters. Multiple sheets of each plastic type are stacked, providing loadings of 0.0254, 0.0508, 0.0762 and 0.1016 g cm$^{-2}$ for the Acetate film (figure 2b) and 0.011, 0.0219, 0.0329, 0.0439 and 0.0548 g cm$^{-2}$ for the Polypropylene plastic.

Grenfell et al. (2011) showed that, for small mass loadings, for small changes in absorbance measured by the integrating sandwich spectrometer the mass loading of the filter and the absorbance measured by the integrating sandwich are linearly





related. In this study, we considered the linear sensitivity between the black carbon mass loading and the black carbon absorbance with the ratio between black carbon and plastic and we equate the ratio of sensitivities to the ratio of the mass absorption cross-section. Therefore, the mass absorption cross-section of the black carbon is expressed in equation 4:

$$\sigma_{BC} = \sigma_{\rho l} \frac{\alpha_{\rho l}}{\beta_{bc}} \tag{4}$$

where $\alpha_{\rho l}$ is the slope of the linear regression between the mass loading of the plastic calibration sheets and the relative absorbance of the plastic measured in the integrating sandwich spectrometer, $\beta_{bc}$ is the slope of the linear regression between the mass loading of the black carbon filters and the relative absorbance of the black carbon measured in the integrating sandwich spectrometer and $\sigma_{\rho l}$ is the mass absorption cross-section of the plastic, given by the Beer-Lambert law.

The mass absorption coefficients of the accetate and polypropylene sheets are measured in a standard spectrometer using
Beer-Lambert law. The measured mass absorption coefficient is $45.77\pm0.04$ cm$^2$g$^{-1}$ ($\lambda$ = 610 nm) for the Acetate plastic and $229.23\pm0.02$ cm$^2$g$^{-1}$ ($\lambda$ = 610 nm) for the Polypropylene plastic.

To visually investigate the size and shape of the black carbon particles used in the experiment, scanning electron microscopy (SEM) is employed. Approximately 6 mm wide squares of the $0.4\mu$m filters containing black carbon particles were cut and glued on standard 12.7 mm diameter SEM stubs using double-faced carbon tabs. The samples were gold coated using a Polaron
E5100 Series II Cool Sputter Coater for 3 minutes in air, creating a 45 nm thick coating. SEM images were generated on a Hitachi S3000N scanning electron microscope. The images were obtained at a magnification of 4000x at a working distance of 12.5 mm, with an acceleration energy of 20 kV and a beam current of 85000 nA. Figure 3 shows a SEM image of black carbon particles on a 0.4 $\mu$m Nuclepore filter. The SEM images are analysed using the ImageJ image analysis software (Abramoff et al., 2004), to determine the size distribution and the circularity of the black carbon particles. The circularity of the particles
is determined by the shape factor Heilbronner and Barrett (2013), caculated using equation 5:

$$SF = \frac{4\pi A}{P^2} \tag{5}$$

Where A is the area of the shape and P, the perimeter of the shape. The shape factor represents the deviation of the perimeter of a particle from a circle of the same area. Values of the shape factor vary between 0, representing an elongated shape and 1, describing a circle. The average shape factor of the particles shown in figure 3 is 0.842, indicating a rough spherical shape.
Assuming a spherical nature of the particles, the diameter is calculated as the maximum Feret diameter. The average diameter of the particles shown in figure 3 is $0.461\pm 0.331$ ($2\sigma$) $\mu$m.



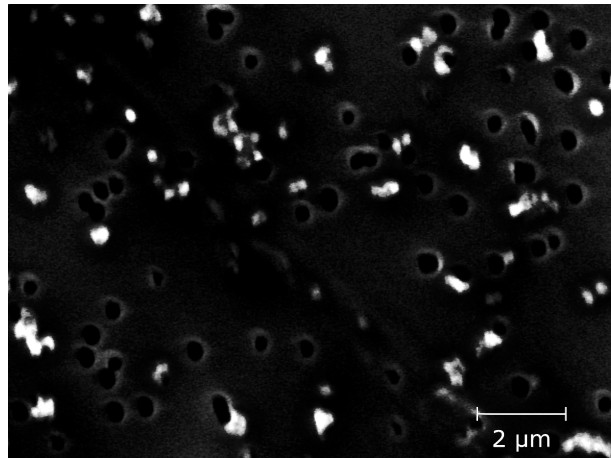

**Figure 3.** Scanning electron microscope image of gold coated black carbon particles (white) at a magnification of x4000, showing a roughly spherical shape of the particles and an average particle diameter of 0.461±0.331 $\mu$m. Note the image also shows the Nuclepore filter holes at 0.4$\mu$m diameter

The mass absorption coefficients is also estimated by a Mie light scattering calculation using the SEM data. Mie calculations are performed using data from the SEM to provide a check of the value for black carbon absorption-cross section for the radiative-transfer calculations. For the Mie calculations the black carbon diameter of 0.461 $\mu$m is used with a density of 1.8 g cm$^{-3}$ and a commonly used refractive index of 1.8 − 0.5 Clarke and Noone (1985), giving an absorption cross section at a wavelength of 550 nm of 2.78 m$^2$g$^{-1}$.

In the work presented here a absorption cross-section value of 2.5 m$^2$g$^{-1}$ will be used for radiative-transfer calculations, as this is between the values from the Mie calculations and the upper limit of values from the integrating sandwich spectrometer.

## 3  TUV-snow radiative-transfer calculations

Calculations using the TUV-snow radiative transfer model (described in section 3.1) are undertaken to simulate optical and physical properties measured of the sea ice. For undoped ice reflectance and $e$-folding depth are calculated (section 3.2) while for sea ice with black carbon the model is used to calculate only reflectance owing to black carbon (section 3.3).

### 3.1  The TUV-snow radiative-transfer model

The TUV-snow model is a coupled atmosphere-snow-sea ice radiative-transfer model using the DISORT code (Stamnes et al., 1988) and is described in detail by Lee-Taylor and Madronich (2002). The model parameterizes sea ice optical properties using only an asymmetry factor, $g$, a wavelength independent scattering cross-section, $\sigma_{scatt}$, a wavelength dependant absorption cross-section, $\sigma_{abs}^+$, and sea ice density and thickness.



## 3.2 Calculations of undoped ice reflectance and $e$-folding depth

The reflectance and $e$-folding depth of the undoped sea ice are calculated through radiative-transfer calculations using the TUV-snow model with derived scattering and absorption cross sections for the ice. To derive these values, values of scattering and absorption cross section are varied until they reproduce the experimentally measured reflectivity and $e$-folding depth data

for the sea ice as detailed in King et al. (2005); France et al. (2011, 2012); Marks and King (2014). Ice density and thickness are measured from ice cored at the end of an experiment. The density of the ice is detailed in table 1. The ice is modelled with a 30 cm thick bottom layer subdivided into 45 sub-layers with each sub-layer representing 1 cm apart from the bottom and top 5 sub-layers which are 1mm thick. The asymmetry factor for the ice is fixed at 0.95, based on a value suggested by Mobley et al. (1998) for a bubble rich ice, which is observed in ice cores taken from the tank.

All calculations are undertaken between wavelengths 350–650 nm, using eight-streams in DISORT. The reflectance under the ice is the measured, wavelength dependent, nadir reflectance of the bottom of the water filled tank. The model illuminates the ice with diffuse light.

Reflectivity is calculated as the ratio of upwelling, $Irr_{up}$, to downwelling $Irr_{down}$, irradiance at the surface of the sea ice, $\left(\frac{Irr_{up}}{Irr_{down}}\right)$. The $e$-folding depth is calculated using equation 2, and the irradiances calculated at depths of 5, 10, 15 and 20 cm

in the sea ice with reference irradiance at a depth of 5 cm (to reproduce experimentally derived $e$-folding depths).

## 3.3 Calculating surface reflectance of ice with a black carbon doped layer

The radiative transfer modelling was repeated for the black carbon doped sea ices. For these radiative transfer calculations parameters are kept the same as the undoped ice calculations, although the ice is modelled as two layers; a 30 cm thick undoped bottom layer and a 5 cm upper, black carbon bearing, layer. These principal layers are subdivided into 45 sub-layers

for the bottom layer and 14 sub-layers in the top layer, with each sub-layer being 1 cm thick, apart from 0.5 cm either side of a boundary (air-ice, undoped ice-doped ice and ice-water), where sub-layers are 1 mm.

The absorption cross-section of the top layer was the same as the bottom undoped layer plus additional absorption for any black carbon present (0, 75, 150 and 300 ng g$^{-1}$), where the absorption cross-section of the black carbon is 2.5 m$^2$g$^{-1}$ (see section 2.3.2). The scattering cross-section of the top layer was derived by a trial and error method to obtain the best fit (judged

by eye) between modelled and measured values of reflectance and $e$-folding depth as described in section 4.3. Densities of the top layers are shown in table 1.

## 4 Results

### 4.1 Physical properties of sea ice and growth rates

Salinity, density and temperature depth profiles of all ice cores are given in the supplementary information. A typical salinity

and temperature profile is given in figure 4. The average density for the top and bottom layer for each black carbon loading is shown in table 1.

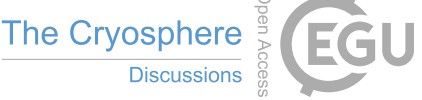



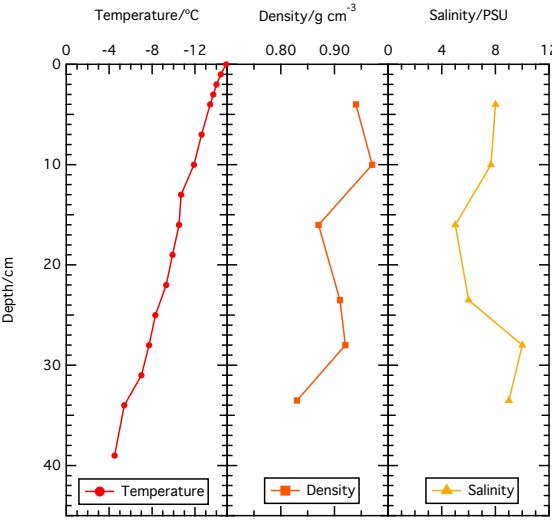

**Figure 4.** Example temperature, salinity and density data for an ice core from run 2 before the black carbon bearing layer was added.

Ice growth rates were similar for all runs with the ice growing at approximately 1.8 cm per day. For all runs the growth rate gradually declined as ice growth progressed. The time taken for reflectance of the ice to become a constant value became longer, taking 5 days for run 1, 7 days for run 2, 11 days for run 3 and 13 days for run 4 owing to an issue with the maintenance of the refrigerator plant which reduced its heat flux but not its maintained temperature. The sea ice produced for each run had

5    a slightly different fabric and subsequently produced less scattering sea ice as shown by the value of $\sigma_{scatt}$ in table 1.

## 4.2    Measured and modelled reflectivity and e-folding depth of undoped sea ice

### 4.2.1    Experimental measurements and calculations of reflectance and e-folding depth for undoped ice

The measured nadir reflectance of the undoped ice layer, is shown in figure 5 for the four runs. Each run represents an experiment with new sea ice growth before the black carbon bearing layer is added. The reflectance is wavelength dependent peaking

10    at values around 500 nm, as would be expected for sea ice (e.g. Grenfell and Maykut (1977)). Measurements of reflectance shown in figure 5 are the average of 5 days of reflectance measurements taken when ice reflectance had become constant. The reflectance of the undoped sea ice decreases from run 1 to run 4 which is attributable to the slightly different ice fabrics in each run and the fact that the ice thickness is not optically thick.



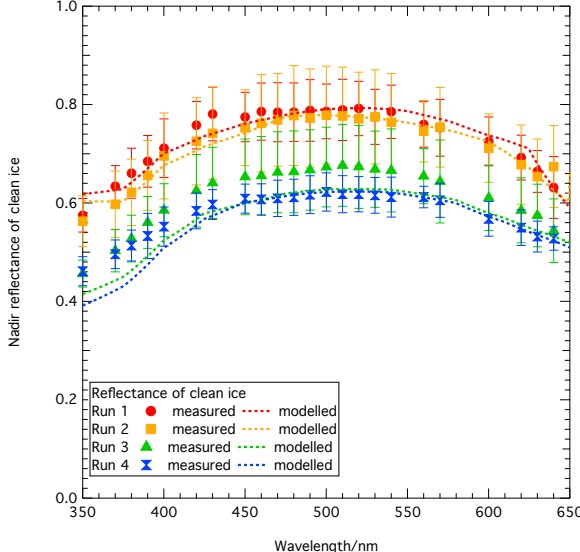

**Figure 5.** Measured sea ice surface reflectance versus wavelength (solid shapes) and modelled sea ice reflectance versus wavelength (dashed lines) for sea ice with no added black carbon

The *e*-folding depth of the undoped ice, figure 6, is also wavelength dependent with the largest values observed around 550 nm for all runs again representing a natural sea ice (e.g. Grenfell and Maykut (1977)). The *e*-folding depths increase with run number which is again attributable to the different ice fabrics created. The increased *e*-folding depth can be attributed to a less light scattering sea ice matrix.

5     Figures 5 and 6 also contain the modelled reflectivity and *e*-folding depth fitted to the experimental data. With the exception of the UV nadir reflectivity of run 3 and 4 the modelled fits are well within uncertainty of the measurement.

It should be noted it is more difficult to find a pair of values for $\sigma_{scatt}$ and $\sigma_{abs}^{+}$ for each wavelength that produce a good reproduction of the experimental reflectivity *and* *e*-folding depth than reflectivity alone as the system described here is significantly more constrained in the degrees of freedom. Thus measuring reflectivity and *e*-folding depth gives the reader more

10   confidence in the values of $\sigma_{scatt}$ and $\sigma_{abs}^{+}$.



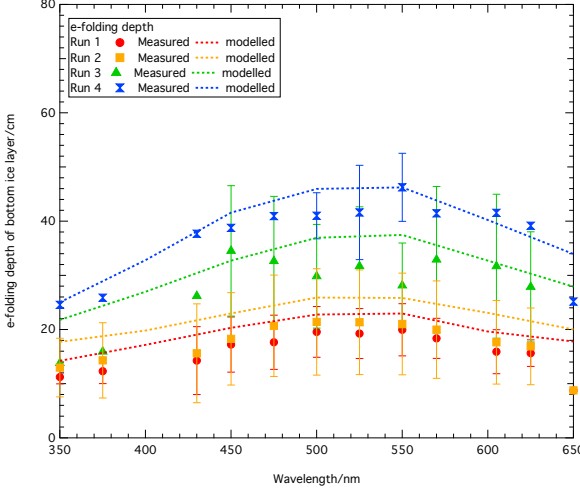

**Figure 6.** Measured sea ice *e*-folding depth versus wavelength (solid shapes) and modelled sea ice *e*-folding depth versus wavelength (dashed lines) for sea ice with no added black carbon.

### 4.2.2 Derived absorption and scattering cross-sections from experimental data for undoped ice

The calculated values of the absorption cross-section of impurities in the undoped ice used in the radiative transfer calculations are shown in figure 7. Ideally this absorption should be zero for undoped ice (no impurities) but a characteristic signal of algae is present (e.g. Bricaud et al. (2004)). The shape of the derived absorption cross-section for each run is similar, decreasing
5   slightly with increased run number.

It should be noted that the algae was unintentional, not observed by the naked eye and resisted several cycles of disinfection. It is testament to the sensitivity of the technique for deriving absorption and scattering cross-sections that the absorption cross-section of the algae can be calculated.



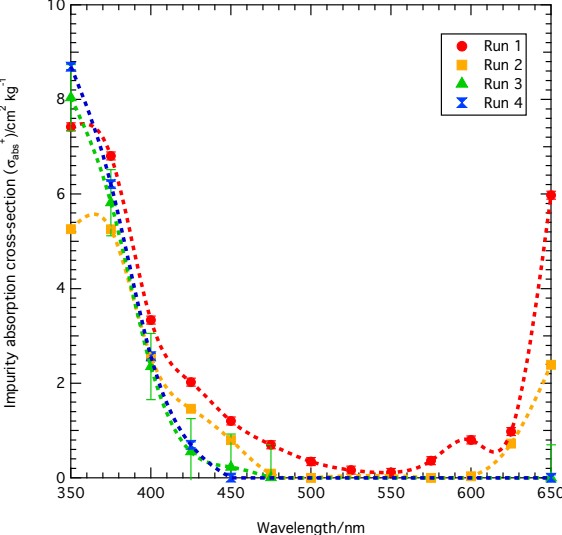

**Figure 7.** Wavelength dependent absorption cross-section derived from reflectance and *e*-folding depth data from runs 1 to 4 for the undoped ice. A smooth line is added to guide the eye. Values for run 4 are too small to plot clearly at longer wavelengths. Error bars show the possible variation in absorption cross-section obtained from different fits of the original reflectance and *e*-folding depth data.

The modelled scattering coefficient used in the radiative-transfer calculations is wavelength independent and reported in table 1.

## 4.3 Surface reflectivity of black carbon doped sea ice

The reflectance of the sea ice with an extra 5 cm black carbon bearing ice layer decreases at all wavelengths as the black carbon mass-ratio increases; as shown in figure 8. At a wavelength of 500 nm, reflectance decreases to 97% of the reflectance of undoped ice (Run 1) for an addition of seawater with a mass ratio of black carbon of 75 ng g$^{-1}$, to 90% for an addition with a mass-ratio of 150 ng g$^{-1}$ compared to undoped ice and to 79% for an addition with a mass ratio of 300 ng g$^{-1}$ compared to reflectivity of undoped ice.





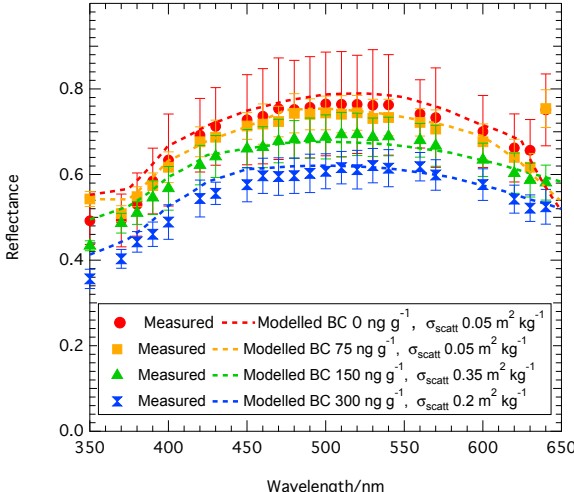

**Figure 8.** Comparison between measured (solid shapes) and calculated (dashed lines) reflectance of simulated sea ice surface with black carbon in a 5 cm surface layer in varying mass ratios. The figure shows the best fit possible by altering scattering cross-section of upper black carbon bearing layer (values shown on plot).

The *e*-folding depth of the ice after the black carbon layer was added was not measured as the total ice could no longer be considered a homogeneous medium and the 5 cm doped layer was too thin to measure the *e*-folding depth for this layer alone.

The best fit obtained between the measured and calculated reflectance values from varying the scattering cross-section of the upper sea ice layer between realistic values is shown in figure 8. These values of the scattering cross-section are shown in figure 8, varying from 0.05 $m^2$ $kg^{-1}$ to 0.35 $m^2$ $kg^{-1}$. The fit between the measured and calculated reflectance is well within uncertainty limits for all runs demonstrating the TUV-snow model can reproduce experimental albedos for sea ice doped with black carbon, even with varying fabrics of sea ice.

## 5 Discussion

In the discussion section possible sources of uncertainty in the experimental measurements compared to the calculated values will be discussed as well as the realism of the simulated sea ice and potential limitations of the sea ice simulator.

### 5.1 Sources of uncertainty in the experimental measurements compared to the calculated values

The comparison between experimentally measured and calculated values reported here are presented under the assumption that the experimental conditions are accurately replicated by the TUV-snow radiative-transfer model. Potential sources for uncertainty in comparing experimentally measured values to calculated values include: aggregation of black carbon particles; mobilisation of black carbon from the top layer of sea ice into the underlying ice and sea water; the value of the asymmetry parameter used in the radiative transfer modelling; uncertainty in the derived scattering and absorption cross-sections of the



experimental sea ice and uncertainty in the mass ratio of black carbon added to the simulated sea ice. The possible contribution of each of these factors is subsequently reviewed.

### 5.1.1 Aggregation of particles

The effect of aggregation of black carbon particles decreases the absorption cross-section in two ways. Assuming two black carbon particles aggregate to form a new spherical black carbon particle, the newly formed particle would have twice the volume and the radius would be $\sqrt[3]{2}$ larger. Mie calculations show that the absorption cross-section would decrease and the mass-absorption cross-section would decrease and flatten according to Dang et al. (2015). Secondly aggregation would reduce the number density of black carbon particles further reducing the absorption of light within the ice. However, the good agreement shown between measured and calculated reflectance for the black carbon doped ice suggests aggregation is not occurring within the ice.

### 5.1.2 Black carbon mobilisation

The radiative-transfer model assumes the black carbon is distributed evenly within the black carbon doped layer, however the experimental ice may not have had an even distribution. Sea ice is at its eutectic point and forms brine pockets and brine channels on freezing (e.g. Weeks (2010)). As the extra layer of seawater freezes brine will drain downwards into the layer of ice below and also be expelled onto the surface. Eicken (2003) suggest that impurities in seawater are trapped in the brine inclusions as sea ice grows. If black carbon is situated in brine inclusions then some of it may drain into the underlying layer of sea ice and eventually into the underlying seawater as described by Eicken (2003). Doherty et al. (2010), who measured black carbon mass-ratios in sea ice in the Arctic, showed that black carbon is concentrated at the surface of the sea ice but also found in smaller concentrations throughout the ice, supporting the idea that black carbon deposited onto the surface of sea ice can be mobilised through the ice. If there was a reduced black carbon mass ratio in the upper layer then surface reflectance would increase (e.g. Marks and King (2013)).

### 5.1.3 Asymmetry parameter

A value of the asymmetry parameter, g, of 0.95 was used based on the work of Mobley et al. (1998). However Mobley et al. (1998) demonstrate that g may vary between 0.94 and 0.99. Figure 9 shows the effect on the determination of the absorption cross-section, $\sigma_{abs}$ owing to absorption by impurities and the ice scattering cross-section at 400 nm for changing the value of $g$ within possible values for sea ice; 0.94–0.99 (Mobley et al., 1998).




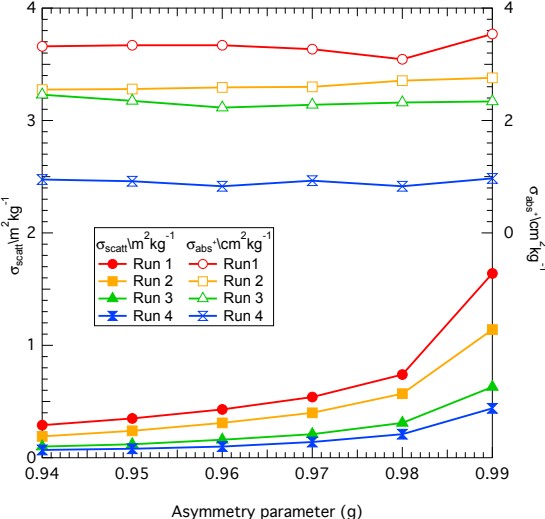

**Figure 9.** Variation in absorption cross-section, $\sigma_{abs}^+$, (cm$^2$ kg$^{-1}$) (dashed line) and scattering cross-section, $\sigma_{scatt}$, (m$^2$ kg$^{-1}$) (solid line) owing to variation in the asymmetry parameter at 400 nm. Note the scatter in the values of $\sigma_{abs}^+$ is due to the fitting process used.

The change in the $g$ value has very little effect on the values of absorption cross-section, as also noted by Libois et al. (2013), with the standard deviation of $\sigma_{abs}^+$ across g = 0.94–0.99 having only a factor of 0.092 effect on the absorption cross-section, demonstrating the model is insensitive to the value of g for determining light absorbing impurities

In the case of scattering cross-section, the effect of changing $g$ from 0.94–0.99 at 400 nm has a much larger effect on the
scattering cross-section, with a larger $g$ value giving a larger scattering cross-section. An increase in the scattering cross-section would change the shape of the reflectance-wavelength curve as well as the values of reflectance, as absorbing impurities have less effect in a sea ice with a large scattering cross-section compared to a small one, as described in Marks and King (2014).

### 5.1.4  Uncertainty in derived scattering and absorption cross-section and black carbon mass ratio

There is a small degree of uncertainty in deriving the values of the scattering and absorption cross-section by modelling from
reflectance and $e$-folding depth measurement data of the undoped ice shown in figure 7 and table 1. The uncertainty was estimated by varying the values of $\sigma_{scatt}$ and $\sigma_{abs}$ and still obtaining a good fit (by eye) to the experimental data.

### 5.2   Realism of artificial sea ice

The sea ice simulator is designed to replicate natural sea ice growth in a controlled environment. Section 5.2 will review how the measured physical and optical parameters of the sea ice compare to field measurements of sea ice to ascertain how the
simulated sea ice compares to natural ice. Although the simulator creates a realistic sea ice environment, as with all simulators, there are limitations in the degree to which a "natural" sea ice environment can be created, limitations in the following metrics were noted; light intensity, uneven ice growth, hyper-saline seawater, surface brine expulsion and reflectance measurements.



### 5.2.1 Physical properties

Temperature profiles from the simulated sea ice show a linear increase in temperature from the surface to the ice base, this has been commonly reported (e.g. Eicken (2003); Perovich et al. (1998b)). Eicken (2003) also suggest that at typical winter temperatures ice would take ∼1 month to form 50 cm, this is a similar growth rate to that observed for the laboratory grown

sea ice where it took approximately three weeks to grow 30 cm of ice.

Typical sea ice densities are reviewed by Timco and Frederking (1996) reporting first year sea ice densities in the range 0.84 to 0.94 g cm$^{-3}$, the density of sea ice created in the simulator ranged from 0.85 to 0.95 g cm$^{-3}$, thus being in the range of natural ice. Perovich et al. (1998b) measured density profiles through Arctic first year sea ice showing no clear variation with depth which is also observed in the simulated sea ice.

Plots of salinity versus depth from ice cores from the sea ice simulator show the distinctive "C" shape with a higher salinity seen at the base and top of the cores, see figure 4. Malgrem et al. (1927) studied salinity of first-year ice also showing a characteristic "C" shape to the sea ice salinity profile. The shape of the salinity profile, explained by Eicken (2003), is due to a combination of salt segregation, gravity drainage and brine expulsion on the surface of the ice. Initially as sea ice in the simulator grows the seawater below the ice increases in salinity and similarly to natural sea ice the seawater below the ice is

commonly hyper-saline. As the simulator continues to generate thicker sea ice there is an increase in the salinity of the brine beneath the sea ice which may eventually retard the growth and the water may become unrealistically saline if the experiment were allowed to continue. In the experiment discussed here this is not a major problem for the experiments as the experiments were performed with ice thicknesses of 30 cm.

The typical structure of a first year sea ice is described by Eicken (2003) showing a granular surface layer, overlying columnar

ice with granular/platelet ice at the ice-water boundary. The structure described by Eicken (2003) is the same as that observed in ice cores of the laboratory grown ice. The surface of the laboratory grown ice has a clear granular texture, and at the base there is a slushy platelet layer with columnar ice in between.

Although the sides of the tank are insulated ice growth across the tank is not quite uniform with slightly thicker ice (∼5 cm) around the edges of the tank towards the end of an experiment (∼3 cm from the sides) and around the polypropylene pole

which the thermocouples were inserted into. Unfortunately the thicker ice areas could not be rectified but are unimportant as reflectance measurements were taken in the same place, away from the sides of the tank.

### 5.2.2 Optical properties

The reflectance of various Arctic sea ice types are reported by Grenfell and Maykut (1977). Figure 10 shows a comparison of the average optically thick reflectance of the laboratory grown sea ice in comparison to the reflectance reported by Grenfell and

Maykut (1977). The reflectance of the laboratory gown ice is considerably larger than a first year ice resembling a reflectance closer to a multi-year ice. The difference may be due to the reflecting surface under the laboratory sea ice being a tank as opposed to a comparatively unreflective ocean. In the radiative-transfer calculations described here the shallow depth of the sea ice tank is accounted for by measurement of its reflectivity.



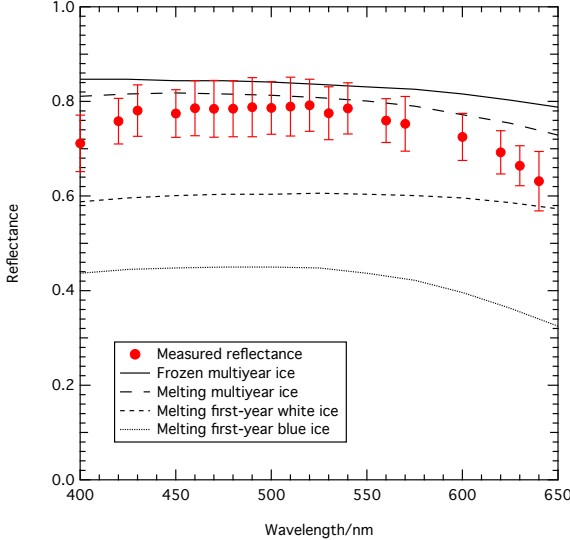

**Figure 10.** Comparison of measured sea ice simulator ice reflectance (red dots) to reflectance values of sea ice measured in the field by Grenfell and Maykut (1977) (black lines).

Typical *e*-folding depths of sea ice at around 500 nm range from ∼1 m for a first year blue ice to ∼35 cm for a multi-year granular white ice (Grenfell and Maykut, 1977). Calculated *e*-folding depths for the laboratory grown ice range from 10–35 cm. The shorter *e*-folding depths calculated for the laboratory grown ice is likely to be due to light reflected from the bottom of the tank and is accounted for in the radiative-transfer modelling.

Overall the sea ice simulator creates a realistic sea ice, recreating typical growth rates, salinity and temperature profiles, reflectance and *e*-folding depths of a first year sea ice.

## 6  Conclusions

The study has shown that the TUV-snow radiative transfer model can reproduce albedo of undoped and black carbon doped sea ice with different sea ice fabrics and thus the model can be used with confidence. Black carbon in simulated sea ice has been shown to reduce the albedo of the ice by 97%, 90%, and 79% compared to clean ice at a wavelength of 500 nm for mass ratios of 75, 150 and 300 ng g$^{-1}$ of black carbon respectively in the top 5 cm layer of the simulated sea ice, which is in agreement with radiative-transfer calculations. To reproduce the albedo using the the TUV-snow model measured albedo and *e*-folding depth data from simulated sea ice is used to derive scattering and absorption cross-sections of the ice using the TUV-snow model. The derived scattering cross-section values are typical of sea ice, while the derived absorption cross-sections show the presence of other absorbing impurities in the undoped ice, which matches the absorption spectra of algae.



*Acknowledgements.* M. Lamare wishes to thank NERC for support from NE/K00770X/1. M. D. King acknowledges support from RHUL for construction of the sea ice simulator with money from JIF funds. All authors thank the Cabot Corporation for the Monarch 120.



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
