# Peer review of "Optical properties of sea ice doped with black carbon- An experimental and radiative-transfer modelling comparison"

_The Cryosphere, 2017_

## Referee Comment (RC1) · Anonymous Referee #1 · 7 Jun 2017

General overall comments:

Positive: The work presented here by Marks et al. is both novel and an excellent fit for The Cryosphere. It is very timely with some of the biogeochemistry sea-ice models currently looking to expand and incorporate a more complex radiative-transfer component for light propagation, and black carbon on sea-ice is topical given the possibility of further opening of Arctic shipping lanes. If the data in here is picked up by the relevant modelling teams I could envisage this being highly citable. The paper also serves as a successful demonstration of the creation of a new sea-ice laboratory, and I would anticipate that this paper will be well utilised in forming the basis for many future experiment

designs.

I have no major concerns over the methodology used, it is clear that the research group have a long history of these kind of experiments in snow and ice and have cited the previous work that this builds on. The use of TUV-snow for the modelling aspects appears to be an entirely valid use of the model given previous publications.

Negative: The abstract feels rushed and confused compared to the rest of the paper and does not do the paper justice.

Much of the modelling community will be put off / not find this paper due to the terminology used. I suggest also quoting e-folding depths as extinction coefficients as these are the terms currently used by the majority of sea-ice models.

The laboratory description feels short and underplayed, this is a new facility and it is very difficult to visualise with the current amount of information provided. I suggest significantly increasing the information provided about the facility, although some could go in the supplementary section. I feel this is important, especially if the authors plan on using this paper as a background reference point to further papers using the facility. Some of the broad comments here are brought up in more detail in the specific comments.

Specific Comments: (Format for reference, e.g 5.4: refers to Page 5, line 4.)

1.3: Question the use of "simulated" throughout, it gives an initial impression of modelling rather than experimental. Could this be altered to laboratory or artificial or something similar?

Abstract General: It is not clear from the abstract what the focus of the paper truly is, and it feels rushed leaving more questions than useful data in its current form. I would suggest removing discussion of algae (as this is barely mentioned in the main manuscript) and refocussing the abstract on the less technical aspects. The mention of measuring e-folding and reflectance and then later calculating e-fold and reflectance

from absorption and scattering values sounds confusing in an abstract and is confusing to the reader. I would remove technical detail in favour of adding extinction coefficents which may be of more value to the readership.

3.9: I would argue that it is a medium sized facility, as somewhere like SERF is a large facility. Could this be phrased in a more impressive way? E.g. "sea-ice simulator designed to reproduce polar sea-ice growth conditions under UV and Visible lighting"

3.14: What is the temperature stability of the cold room?

4.Fig1. Is there wind shear across the tank? It'd be good to have it added to the Figure.

4.General. It'd be really nice to see the facility description fleshed out more, with some more technical details of what the chamber is capable of (especially as this is appears to be the first paper to come out from the laboratory). Some suggestions would be how temp / salinity of the ocean vary through time of an experiment as the ice grows, lighting consistency at the ice surface, room temperature vs time during ice growth. These are just suggestions, but it would be good to have more technical facility description. Could some annotated photos be added to go alongside the figures? This would not only allow a much better visualisation, but could well encourage potential collaborators.

5.14: Model # for the thermocouples? Details on precision / calibration if possible?

6.7: Again, could an annotated photo be added for the set up or an extra figure? It may not be implicit for non experimentalists to imagine.

7.7: Clarify, is the tank emptied at this point or is bleach added?

7.27. Can the authors clarify whether this is a range, or the result of two experiments? Is there any way of knowing where the differences in result occur from?

8.Table 1: No mention of sigmascatt in the caption. Please add details.

9. Figure 2. Caption is not sufficient and needs fully re-composing.

11. Fig 3: It could be due to black and white printing, but the contrast seems very off. It is a really nice Figure to have, but it currently is not as clear as it could be.

11.5: The authors should be commended here for using a secondary method to calculate an important parameter for the work. This sort of thing is often overlooked and should be done much more in many fields of science.

14.3 Is there any way of further discussing the fabrics of the ice? Is there any information in the cores that could be used? Currently the images in the SI are too small to really ascertain anything structural but maybe there is information within those images which could be enhanced to help interpret the variability?

15.Fig 7: Please add a scaled absorption of algae (and what type of algae) be added here.

17.16: (and 19.10:) How does the uncertainty in the e-fold fitting procedure propagate through? There is much discussion about the other parameters but I feel that this is overlooked and that there are sources of error which are not propagated from the experimental side? Is there an easy way to estimate this?

20.25: Would the extra ice at the side have any impact on the e-folding depth?

21.4: It is my feeling that if light is being reflected back from the base of the tank, then there would be excess light within the ice at depth, which would result in the e-folding depth becoming longer and not shorter.

I would also disagree with the authors comparison to the blue ice seen in Grenfell and Maykut (1977) as this is described as "ice saturated, but not covered, with melt water". The ice created here is fresh, "dry" ice and has not been subjected to melt metamorphism and structural change such as the one described in Grenfell and Maykut 1977, and should not be compared as such.

Technical Comments: 2.26 Unclear, are these the first experiments or just the first experiments using TUV-snow? 2.30 Personal preference, and may be disagreed by

the editorial team, but this list of aims feels very wordy. Could it be bulleted? 7.13 Extra space needed. 9.8: Sentence currently doesn't make sense, too many "for smalls"? 10.9 "with a", not in a?

---

## Short Comment (SC1) · 12 Jun 2017

S. Doherty

sarahd@atmos.washington.edu

A (likely) correction to the Abstract: "Particulate black carbon at mass ratios of 75, 150 and 300 ng/g in a 5 cm ice layer lowers the albedo by 97 %, 90 %, and 79 % compared to clean ice at a wavelength of 500 nm." I believe that the authors mean that it "lowers the albedo to (i.e. not "by") 97%, 90% and 79%" – or some other wording adjustment is needed. Lowering it *by* 97% would make for some very black ice indeed!

---

## Referee Comment (RC2) · Anonymous Referee #2 · 15 Jun 2017

This manuscript describes results from a laboratory experiment and related numerical modeling. The experiment entails the growth of laboratory-simulated sea ice. Four ice blocks were grown, the first with no black carbon (BC) added, the following three each with progressively larger concentrations of BC. The BC was restricted to a 5cm thick surface layer. As each ice block was grown, spectral reflectance was recorded. At the end of each growth, the vertical profile of upwelling radiance within the ice was measured. The manuscript then describes the comparison between observed apparent optical properties and apparent optical properties predicted with the "TUV-snow" radiative transfer model. The overall topic of this manuscript is of interest to the TC readership. The results of the paper are interesting and timely. However, I have a num-

ber of both major and minor concerns that need to be addressed before this paper can be published.

Major concerns: 1. The paper is not particularly well written. I find numerous instances where the writing is sloppy and imprecise. I will attempt to point them out in my minor comments below, but the manuscript could be dramatically improved if some attention were given to the writing.

2. There seems to be a mismatch between the title and the stated objectives. The title suggests the point of the paper is to present the optical properties of laboratory-grown sea ice containing black carbon. The abstract, however, starts by suggesting the reader should expect a manuscript detailing the validation of a radiative transfer model. P. 1 lines 1-4 are really not clear what this manuscript is setting out to do.

3. I am not entirely comfortable with the nature of the comparisons being made between the observations and the model. There seems to be some circularity here. In the abstract (lines 6 – 7) it is stated that measured apparent optical properties (albedo and extinction) are used to derive inherent optical properties (scattering and absorption cross-sections) "using the model". It is not at all clear what this means. Then lines 10 -12 state that light extinction (e-folding depth) is calculated using the model and the IOPs that were derived directly from AOPs (lines 6 -7)? This sounds rather circular—-like saying that the measurements are used to define the inherent optical properties of the domain (using the model), which are then fed back into the model to produce apparent optical properties, for comparison with the measured AOPs. Well, I would hope those would agree! Page 2 line 30 states that it is the third objective of this work to use measured [apparent] optical properties to recreate the irradiance within the sea ice using the TUV-snow radiative transfer model and compare modelled and measured values. To me, this says that the objective is to use the observations to infer IOPs appropriate for input for the model, and to then compare modeled and measured AOPs. I don't think this is a legitimate comparison! The model is being forced to agree with the observations! There is no independent comparison here. The further discussion on p.

[Figure]

12 (lines 2 – 4) reinforces this circularity.

4. Use of upwelling radiance to determine e-folding depth in finite-depth domain with forward peaked scattering phase function? If I understand correctly, the e-folding depth is calculated from the measurement of upwelling radiance. I would expect the measurement of e-folding depth in this relatively thin (30 – 50 cm) ice block to be biased low, but measuring the upwelling radiance makes it only worse. Take the limiting case of an upwelling radiance measurement immediately above the ice-water interface. I would expect it to be near zero, whereas the downwelling radiance would be non-zero. The e-folding depth from those two different measurements should be quite different.

5. Confusion between radiance and irradiance? I thought that the ratio of upwelling nadir radiance to downwelling nadir radiance was being measured, but on page 12 line 13 it sounds like the measurement is ratio of upwelling irradiance to downwelling irradiance. Is there confusion here between "radiance" and "irradiance"? They are radiometrically distinct quantities and should not be interchanged. Additionally, p. 20 line 30 states that the reflectance of the laboratory grown (not 'gown') ice is considerably larger than first year ice, and resembles a reflectance closer to multi-year ice. Does this statement mean that the spectral ratio of upwelling to downwelling radiance (as far as I can tell, the only optical property measured above the ice) is being compared to spectral albedos published in the literature? Here again, I think it is possible there may be some confusion between radiance and irradiance? Or is the model being used to estimate the albedo of the ice—which is then being compared to the albedos of natural ice? What natural ice measurements are being used in this comparison? Also, it would be helpful to present the measured spectral reflectivity of the tank, since it possibly matters so much.

6. Figure 7 shows wavelength-dependent absorption cross-section derived from reflectance and de-folding depth data from the four runs, with no BC. I am concerned about the interpretation of these data. These curves don't really look like chlorophyll absorption spectra to me. Chlorophyll typically has absorption maxima at 430 – 450

nm and 640 - 670 nm. How was the absorption of water and ice represented in the model? Is it possible there was some error in representing the ice and/or the brine, and these spectra, which look similar in nature to the absorption of water?

Minor points: P. 1 L 14: As pointed out by a different reviewer in a short comment, the last line of the abstract states that "albedo is reduced by" as much as 97%. This cannot be accurate!

p. 1 L 22: What is the "TUV-snow radiative transfer model for sea ice"? I am not familiar with it, and I find it rather confusing that it is a "snow" model for "sea ice". What does TUV stand for?

p.2 L2: Here the authors mention that the sea ice simulator has "not been experimentally validated for sea ice." If that is an objective of this manuscript then it should appear perhaps in the title, and probably in the abstract. The paper is a bit diffuse because it seems to have many different objectives, as listed at the bottom of p. 2.

p.2 L 14-15: It is not clear why validating the TUV-snow model just for a single type of sea ice, grown under particular circumstances, and a single absorber, in this case BC, necessarily means the model can be used "confidently" for other sea ice types and absorbers. For instance, I can imagine that ice grown with very few scatterers could have much smaller optical depth, and perhaps would be a different modeling problem than the one examined here.

p.3 L 9: "temperature" is "higher/lower", not "warmer/cooler" Figure 1: I see the tank volume is 2000 liters, but there is no indication of the diameter and depth? They matter, particularly in regards to the exchange of salt between the growing ice sheet and the "ocean".

p.4 L7: does the pump achieve vertical mixing? Are you only worried about temperature stratification? What about salinity stratification?

p.4 L14: what does "majority of shortwave solar wavelengths" mean? Please clarify.

p. 4 L18: Was the incident light field isotropic? Or just diffuse? It is difficult to create an isotropic light field in the laboratory, but it is also difficult to simulate a diffuse light field that is not isotropic in a model. Just saying they are both diffuse, does not ensure a valid comparison.

p.5 L7: Drilling a hole breaks the horizontal homogeneity of the ice block, could cause additional brine drainage, and does reduce the integrity of the ice, but the authors should be wary of stating that it "destroys the fabric of the ice", as I don't think this is accurate.

p.6 L5: what size is the reflector panel? At some size it will reflect significant radiation back to the "sky" (lighting panels and white boards) and enhance the downwelling radiation field, biasing the reflectance. Please state the size of the panel and discuss the possibility of it affecting the measurement of the incident light.

p.6 L22: what does it mean that the "e-folding depth . . . is asymptotic"? please clarify. Also, it is not accurate to say "there are no near surface effects". The fact that this is a finite domain means there necessarily will be some surface effects.

p.7 L17: "proportion"? I think "portion" is intended?

p.7 L27: why two (very different!) values for the mass absorption cross-section?

Table 1: units for density are not g cmˆ3. Also, I am confused about the cross-section units of cmˆ2 kgˆ-1. Cross-sections on previous page are cited in mˆ2 gˆ-1. Those are not equivalent.

Fig. 2 Y-axes have different labels—should they not both be "Relative spectral absorbance"? I understand the two figures are for different materials, but I think they are intended to be compared, and if that is so then they should have the same label on their y-axes.

p. 9 L 8 and following. This sentence is cryptic. It needs to be rewritten for clarity.
p. 12 L 5: Why is ice density measured and reported? Is it used in the modeling? If so, the way that is used could be important and should be described.

p. 12 L 10 -11: Sentence beginning "The reflectance under..." needs to be rewritten for clarity.

p. 13 L 4- 5: Higher air temperature should result in slower ice growth. Slower ice growth would be expected to result in less entrainment of brine within the ice. Less brine would be expected to yield fewer and/or smaller brine inclusions, which would then result in reduced scattering.

p. 14 L 3 -4: see comments above about reduced salinity The data displayed in figures 5 and 8 really should be presented on the same plot; it is very difficult to make the comparison when they are in different figures.

p. 18 L 13: No, sea ice is not at its eutectic point, unless it is very cold (about -37 C). When in thermal equilibrium, it is always at its melting point, hence the required equilibrium concentrations of brine and ice.

Figure 9: please specify which y-axis corresponds with which curves.

p. 19 L 4 -7 : This relates to a commonly recognized "similarity principle" in radiative transfer.

p. 20 L 15, following: does the exchange of sea water in the "ocean" of the simulator correct for salinity variations? I would expect even a 30 cm thick ice cover could affect the salinity of the ocean, but since the dimensions of the tank are not given (other than total volume), it is impossible to estimate the salinity enhancement in the 'ocean' due to freezing of the ice and resulting salt rejection.

p. 21 L 11: Here again, "...reduce the albedo of the ice by 97%..." I think this should be "...reduce... to 97%".

p. 21 L 15: "The derived scattering cross-section values are typical of sea ice..." what

are the derived values being compared to?

---

## Author Comment (AC1) · 25 Aug 2017

The authors would like to thank the referee for their review of the paper "Optical properties of laboratory grown sea ice doped with light absorbing impurities (black carbon)."

Changes made to the paper based on the comments are detailed below on a point-by-point basis:

**The abstract feels rushed and confused compared to the rest of the paper and does not do the paper justice.**
The abstract has been rewritten:

[Figure]

*"Radiative-transfer calculations of the light reflectivity and extinction coefficient in laboratory generated sea ice doped with and without black carbon demonstrate that the radiative transfer model TUV-snow can be used to predict the light reflectance and extinction coefficient of sea ice typical of first year sea ice containing typical amounts of black carbon and other light absorbing impurities. The experiments give confidence in the application of the model to predict albedo of other sea ice fabrics.*

*Sea ices, $\sim$30 cm thick, were generated in the Royal Holloway Sea Ice Simulator ($\sim$2000 L tanks) with scattering cross-sections measured between 0.012 and 0.032 $m^2$ $kg^{-1}$ for four ices. Sea ices were generated with and without $\sim$5 cm upper layers containing particulate black carbon. Nadir reflectances between 0.60 and 0.78 where measured along with extinction coefficients of 0.1 to 0.03 $cm^{-1}$ (e-folding depths of 10–30 cm) at a wavelength of 500 nm. Values were measured between light wavelengths of 350 and 650 nm. The sea ices generated in the Royal Holloway Sea Ice Simulator were found to be representative of natural sea ices.*

*Particulate black carbon at mass ratios of $\sim$75, $\sim$150 and $\sim$300 ng $g^{-1}$ in a 5 cm ice layer lowers the albedo to 97%, 90%, and 79% of the reflectivity of an undoped "clean'" sea (at a wavelength of 500 nm)."*

**Much of the modelling community will be put off / not find this paper due to the terminology used. I suggest also quoting e-folding depths as extinction coefficients as these are the terms currently used by the majority of sea-ice models.**

Throughout the paper "extinction coefficients" are now referred to in addition to "e-folding depths" (for the "snow" community) and the following explanation has been added to the text: *"At the completion of the experiment the extinction coefficient and e-folding depth are measured. The e-folding depth is the distance over which light intensity reduces to 1/e of its initial value and is the reciprocal of the extinction coefficient. The e-folding depth is reported in addition to the extinction coefficient."*

**The laboratory description feels short and underplayed, this is a new facility and**

**it is very difficult to visualise with the current amount of information provided. I suggest significantly increasing the information provided about the facility, although some could go in the supplementary section. I feel this is important, especially if the authors plan on using this paper as a background reference point to further papers using the facility. Some of the broad comments here are brought up in more detail in the specific comments.**

The laboratory description had now been expanded to include further details of the facility including a further annotated figure of the facility, and two figures demonstrating the capabilities of the sea ice simulator (temperature profiles and daily reflectance measurements) and further technical specifications. These changes are described in more detail in the specific comments below.

**Specific Comments: (Format for reference, e.g 5.4: refers to Page 5, line 4.)**

**1.3: Question the use of "simulated" throughout, it gives an initial impression of modelling rather than experimental. Could this be altered to laboratory or artificial or something similar?**

The word "simulated" has now been replaced with "laboratory" throughout the manuscript.

**Abstract General: It is not clear from the abstract what the focus of the paper truly is, and it feels rushed leaving more questions than useful data in its current form. I would suggest removing discussion of algae (as this is barely mentioned in the main manuscript) and refocussing the abstract on the less technical aspects. The mention of measuring e-folding and reflectance and then later calculating e-fold and reflectance from absorption and scattering values sounds confusing in an abstract and is confusing to the reader. I would remove technical detail in favour of adding extinction coefficients which may be of more value to the readership.**

The abstract has been completely rewritten. The discussion of algae has been removed from the abstract and the level of technical detail lowered.

**3.9: I would argue that it is a medium sized facility, as somewhere like SERF is a large facility. Could this be phrased in a more impressive way? E.g. "sea-ice simulator designed to reproduce polar sea-ice growth conditions under UV and Visible lighting"**

The phrase: *"The sea ice simulator is a large scale, UK based, laboratory sea ice tank designed to replicate warmer polar temperatures, the ocean and UV and visible wavelengths of solar radiation."* Has been replaced with: *"The sea ice simulator is designed to replicate a Polar sea ice growth environment under UV and visible wavelengths of solar radiation"*

**3.14: What is the temperature stability of the cold room?**

The following sentence has been added to the text: *"The air temperature within the container varies by $\pm 1°C$ although thermocouples monitoring temperature at the ice surface show better temperature stability, whilst the temperature variation measured within the ice is less than the precision of the probes ($\pm 0.2°C$). Every 12 hours the chiller removes ice build-up on the cooling plant and the air temperature rises briefly by $\sim 6°C$."*

**4.Fig1. Is there wind shear across the tank? It'd be good to have it added to the Figure.**

A 20" fan located above the sea ice directed 110 m$^3$ min$^{-1}$ of room air onto the ice at an angle of $\sim 45°$. The air velocity across the surface of the ice was $\sim 1.5$ ms$^{-1}$. The velocity of the airflow produced by the fan has been added to figure 1.

The following text has also been added to the manuscript: *"An additional air fan, attached to the ceiling, blows cold, ambient air at the water surface, ($\sim 100$ m$^3$ min$^{-1}$), increasing the heat flux from the ice surface, quickening ice formation and assisting the*

*production of columnar ice (Weeks, 2010)."*

**4.General. It'd be really nice to see the facility description fleshed out more, with some more technical details of what the chamber is capable of (especially as this is appears to be the first paper to come out from the laboratory). Some suggestions would be how temp / salinity of the ocean vary through time of an experiment as the ice grows, lighting consistency at the ice surface, room temperature vs time during ice growth. These are just suggestions, but it would be good to have more technical facility description. Could some annotated photos be added to go alongside the figures? This would not only allow a much better visualisation, but could well encourage potential collaborators.**

Further technical details have now been added to the laboratory description, including:

An annotated photo of the facility (attached here as figure 1)

A further two figures (new figures 2 and 3, also attached here as figures 2 and 3) have been added to section 2.1 to demonstrate the facilities capabilities. Figure 2 shows the change in ice and water temperature profiles during ice growth. Figure 3 shows the change in ice reflectance during ice growth and the day-to-day reflectance stability of the optically thick ice.

The following additional pieces of text have been added to section 4:

*"Figure 2 demonstrates that the temperature of the water beneath the sea ice is not thermally stratified, sea ice growth is from the surface downwards, ice temperature decreases linearly through the ice with depth and the ice surface temperature is at a constant $-15°C$"*

*"The short term variability of the lamps was less than 0.1% (after an initial warm-up) on the timescale of the measurement of reflectivity or e-folding depth. Note that both the value of nadir reflectance (relative to a Spectralon panel) and light penetration depth are not dependent on the illumination irradiance providing the irradiance does not*

*change during the measurement. Figure 3 shows the change in nadir ice reflectance during ice growth and the day-to-day reflectance stability of the optically thick ice."*

**5.14: Model for the thermocouples? Details on precision / calibration if possible?**

The following sentence has been added to this section: *"The precision on all the thermocouples at $-15°C$ was measured as $\pm0.2°C$"*

**6.7: Again, could an annotated photo be added for the set up or an extra figure? It may not be implicit for non-experimentalists to imagine.**

An extra, annotated, photograph has been added to this section as part of figure 1 (attached here as figure 1).

**7.7: Clarify, is the tank emptied at this point or is bleach added?**

The sentences *"Between experimental runs the tank is periodically bleached to remove any algae that may have grown. No algae was visible to the naked eye"*, has been changed for clarity to read *"The sea ice was melted and the resulting seawater was treated with aqueous hypochlorous acid (HOCl) and filtered between experimental runs to remove any algae that may have grown. No algae was visible to the naked eye"*

**7.27. Can the authors clarify whether this is a range, or the result of two experiments? Is there any way of knowing where the differences in result occur from?**

The value of 0.58 has been removed from the text and was an erroneous value left in from a previous edit.

**8.Table 1: No mention of sigmascatt in the caption. Please add details.**

The caption has been changed to include the sigmascatt, the caption now reads: *"Table 1: Optical and physical properties of sea ice for each run including the mass-ratio of black carbon added to the top layer of ice, density of ice and scattering cross-section*

*(σscatt) of both the top and bottom layers of the ice. The uncertainty….”*

**9. Figure 2. Caption is not sufficient and needs fully re-composing.**

The caption has now been changed to read *"Figure 2: a) Relative spectral absorbance of black carbon versus wavelength for various loadings of black carbon on the fil-ter. b) Relative spectral absorbance versus wavelength for different mass loadings of polypropylene"*

**11. Fig 3: It could be due to black and white printing, but the contrast seems very off. It is a really nice Figure to have, but it currently is not as clear as it could be.**

Figure 3 (now figure 5) has been improved, previously it displayed well electronically as a pdf, but did not print well, so the image has been sharpened and annotated to make the black carbon particles clearer (attached here as figure 4).

**11.5: The authors should be commended here for using a secondary method to calculate an important parameter for the work. This sort of thing is often overlooked and should be done much more in many fields of science.**

Thank you

**14.3 Is there any way of further discussing the fabrics of the ice? Is there any information in the cores that could be used? Currently the images in the SI are too small to really ascertain anything structural but maybe there is information within those images, which could be enhanced to help interpret the variability?**

It has not been possible to enhance the images of the ice fabric any further and the sample no longer exist as they were melted for density measurements.

**15.Fig 7: Please add a scaled absorption of algae (and what type of algae) be added here.** An absorption cross-section for chlorophyll-A from Bricaud et al. (2004) from algal cells, and chlorophyll in ice from Mundy et al (2011) has been added to this figure to more clearly demonstrate the identity of the extra absorption (attached here

as figure 5).

**17.16: (and 19.10:) How does the uncertainty in the e-fold fitting procedure propagate through? There is much discussion about the other parameters but I feel that this is overlooked and that there are sources of error which are not propagated from the experimental side? Is there an easy way to estimate this?**

Section 5.1.4 "Uncertainty in derived scattering and absorption cross-section and black carbon mass ratio" has been altered to read:

*"Section 5.1.4 Uncertainty in derived scattering and absorption cross-section and black carbon mass ratio*

*The determination of the cross-section for light scattering and absorption, described in section 4.2.2, depends on varying their values to reproduce the measured values of the e-folding depth and the nadir reflectivity within the experimental uncertainties of measured values of the e-folding depth and the nadir reflectivity, all as a function of wavelength. The latter assists in constraining the determination of the values of the cross-section for absorption and scattering. The propagated uncertainty in the determined values of the cross-sections for light scattering and absorption from uncertainties in either the value of the e-folding depth or nadir reflectivity in isolation have not been calculated as our method fits both e-folding depth and reflectivity. Considering the experimental uncertainty in e-folding depth and nadir reflectivity gives a more representative uncertainty of the process. The uncertainty in the reflectance and e-folding depth measurement data of the undoped ice is shown in figures 7 and 8. Table 1 gives an indication of the uncertainty in the derived scattering cross-section which is estimated by varying the values of $\sigma$scatt and $\sigma$abs and still obtaining a good fit (by eye) to the experimental data within the uncertainties of the measured e-folding depth and nadir reflectivity.*

**20.25: Would the extra ice at the side have any impact on the e-folding depth?**
The *e*-folding depth is measured more than three *e*-folding depths away from the sides

of the tank so the extra ice would have no effect on measured *e*-folding depth values. We have added the following comment to the text *"All measurements of the e-folding depth were made more than three e-folding depths from the sides of the tank so that any extra ice growth at the edges of the tank would no impact on the measurements."*

**21.4: It is my feeling that if light is being reflected back from the base of the tank, then there would be excess light within the ice at depth, which would result in the e-folding depth becoming longer and not shorter. I would also disagree with the authors comparison to the blue ice seen in Grenfell and Maykut (1977) as this is described as "ice saturated, but not covered, with melt water". The ice created here is fresh, "dry" ice and has not been subjected to melt metamorphism and structural change such as the one described in Grenfell and Maykut 1977, and should not be compared as such.**

The reference to the first year blue ice has been removed from the text, as has the explanation for the shorter e-folding depths being attributed to light reflected off the bottom of the tank. The paragraph now reads: *"Typical extinction coefficients of sea ice at around 500 nm are around 0.03 cm$^{-1}$ (Grenfell,1977). Calculated extinction coefficients for the laboratory grown ice range from 0.1–0.03 cm$^{-1}$."*

**Technical Comments:**

**2.26 Unclear, are these the first experiments or just the first experiments using TUV-snow?**

The sentence *"The study presented here are the first experiments with the Royal Holloway Sea Ice Simulator to evaluate the TUV snow model for undoped sea ice...."* has been changed for clarity to read *"The study presented here includes the first experiments with the Royal Holloway Sea Ice Simulator, the first experiments to evaluate the TUV snow model for undoped sea ice, the first experimental results to demonstrate the change in reflectance owing to light absorbing impurities in sea ice and finally the first experiments to evaluate the TUV-snow model for reflectivity calculations for light*

*absorbing impurities in sea ice. "*

**2.30 Personal preference, and may be disagreed by the editorial team, but this list of aims feels very wordy. Could it be bulleted?**

The list of aims has now been removed following similar comments also from the second referee.

**7.13 Extra space needed.**

An extra space has been added.

**9.8: Sentence currently doesn't make sense, too many "for smalls"?**

The sentence has been changed to read *"Grenfell et al. (2011) showed that for small amounts of black carbon the mass loading is directly proportional to the absorbance measured by the integrating sandwich spectrometer."*

**10.9 "with a", not "in a"?**

Agreed, "in a" has been changed to "with a"
* * *
[Figure]

**Fig. 1.** Updated figure 1 now including annotated photo of sea ice facility

**Reflectance** (y-axis)

**Days of ice growth** (x-axis)

**Fig. 2.** New figure showing daily reflectance measurements during ice growth

[Figure]

**Fig. 3.** New figure showing ice temperature profile measurements during ice growth

Black carbon particle

Nataplore filter hole

Nuclepore filter

2 μm

**Fig. 4.** Updated figure of EM image of black carbon particles now with annotations

[Figure]

**Fig. 5.** Updated figure of ice absorption cross-section now including absorption cross-section of chlorophyll

---

## Author Comment (AC2) · 25 Aug 2017

The authors would like to thank the referee for their comments on the paper "Optical properties of laboratory grown sea ice doped with light absorbing impurities (black carbon).

Changes made to the paper based on their review are detailed below on a point-by-point basis:

Major concerns:

**1. The paper is not particularly well written. I find numerous instances where**

[Figure]

**the writing is sloppy and imprecise. I will attempt to point them out in my minor comments below, but the manuscript could be dramatically improved if some attention were given to the writing.**

The paper has been reviewed to sharpen-up the writing and improve the manuscript.

**2. There seems to be a mismatch between the title and the stated objectives. The title suggests the point of the paper is to present the optical properties of laboratory grown sea ice containing black carbon. The abstract, however, starts by suggesting the reader should expect a manuscript detailing the validation of a radiative transfer model. P. 1 lines 1-4 are really not clear what this manuscript is setting out to do.**

The title of the paper has been changed to include radiative-transfer modelling and now reads: *"Optical properties of sea ice doped with black carbon- An experimental and radiative-transfer modelling comparison".*

The abstract has also been completely rewritten:

*Radiative-transfer calculations of the light reflectivity and extinction coefficient in laboratory generated sea ice doped with and without black carbon demonstrate that the radiative transfer model TUV-snow can be used to predict the light reflectance and extinction coefficient of sea ice typical of first year sea ice containing typical amounts of black carbon and other light absorbing impurities. The experiments give confidence in the application of the model to predict albedo of other sea ice fabrics.*

*Sea ices, $\sim$30 cm thick, were generated in the Royal Holloway Sea Ice Simulator ($\sim$2000 L tanks) with scattering cross-sections measured between 0.012 and 0.032 $m^2$ $kg^{-1}$ for four ices. Sea ices were generated with and without $\sim$5 cm upper layers containing particulate black carbon. Nadir reflectances between 0.60 and 0.78 where measured along with extinction coefficients of 0.1 to 0.03 $cm^{-1}$ (e-folding depths of 10–30 cm) at a wavelength of 500 nm. Values were measured between light*

*wavelengths of 350 and 650 nm. The sea ices generated in the Royal Holloway Sea Ice Simulator were found to be representative of natural sea ices.*

*Particulate black carbon at mass ratios of ~75, ~150 and ~300 ng g$^{-1}$ in a 5 cm ice layer lowers the albedo to 97%, 90%, and 79% of the reflectivity of an undoped "clean" sea ice (at a wavelength of 500 nm).*

**3. I am not entirely comfortable with the nature of the comparisons being made between the observations and the model. There seems to be some circularity here. In the abstract (lines 6 – 7) it is stated that measured apparent optical properties (albedo and extinction) are used to derive inherent optical properties (scattering and absorption cross-sections) "using the model". It is not at all clear what this means. Then lines 10 -12 state that light extinction (e-folding depth) is calculated using the model and the IOPs that were derived directly from AOPs (lines 6 -7)? This sounds rather circular- like saying that the measurements are used to define the inherent optical properties of the domain (using the model), which are then fed back into the model to produce apparent optical properties, for comparison with the measured AOPs. Well, I would hope those would agree!**

The abstract has been re-written to be clearer

**Page 2 line 30 states that it is the third objective of this work to use measured [apparent] optical properties to recreate the irradiance within the sea ice using the TUV-snow radiative transfer model and compare modelled and measured values. To me, this says that the objective is to use the observations to infer IOPs appropriate for input for the model, and to then compare modelled and measured AOPs. I don't think this is a legitimate comparison! The model is being forced to agree with the observations! There is no independent comparison here. The further discussion on p. C2 12 (lines 2 – 4) reinforces this circularity.**

The objectives have been removed as they caused confusion. Also there is no circularity in the procedure presented here; we are demonstrating we can reproduce our

experimental data with modelled data and then perturb the experiment (with black car-bon) and continue to reproduce the experimental data with the model. Furthermore using the reviewer's (and Mobley's) nomenclature:

- The AOPs of the sea ice are measured and modelled to determine values of the IOPs

- To determine how well the model AOPs fit the measured AOPs we compare the modelled AOPs with the measured AOPs

- We perturb the experiment with the addition of black carbon and measure AOPs. We than compare modelled and measured AOPs

Thus we demonstrate that:

1) We can reproduce our experimental AOPs (n.b. we use e-folding depth and albedo, most studies only use albedo thus our system is more constrained) by modelling IOPs of pure ice. The TUV-snow radiative-transfer model can be used to describe the radiative-transfer in and above the sea ice

2) We can perturb the experimental system with a known amount of a known absorber and reproduce the experimental results with our model. Note the system is constrained by amount, size and MAC of an absorber and these cannot be used as modelling parameters such as those used in other studies.

**4. Use of upwelling radiance to determine e-folding depth in finite-depth do-main with forward peaked scattering phase function? If I understand correctly, the e-folding depth is calculated from the measurement of upwelling radiance. I would expect the measurement of e-folding depth in this relatively thin (30–50 cm) ice block to be biased low, but measuring the upwelling radiance makes it only worse. Take the limiting case of an upwelling radiance measurement imme-diately above the ice-water interface. I would expect it to be near zero, whereas the downwelling radiance would be non-zero. The e-folding depth from those two different measurements should be quite different.**

The following text has been added to the manuscript to clarify this: *"At each depth drilled the same fibre optic is inserted into the hole and the light intensity (upwelling radiance) measured via an Ocean Optics spectrometer. In an optically thick sea ice the measurement of either up or downwelling light for e-folding depth is not important as has been shown by France et al. 2012."*

And: *"Measurements used to calculate the e-folding depth are only conducted in the middle of the ice as the irradiance profile changes rapidly at the air-ice and ice-water boundary (a good example shown in King et al. 2005). The calculation of an e-folding depth from the modelled downwelling irradiance was calculated from similar depths as the experimental ice. The modelled ice had the same thickness and underlying tank radiance field as the experiment"*

The authors are well aware of the boundary effects that occur at the air-ice and ice-water interfaces, therefore no measurements were performed near the upper or lower ice boundaries, as we (King, et al 2005) have previously shown the irradiance profile changes rapidly at these edges. The optically "thin" sea ice would not produce an asymptotic value of the e-folding depth in the experiments measured but produce a phenomenological value for this particular experiment. Thus our use of radiative-transfer modelling to reproduce the reflectance and the e-folding depth to determine values of $\sigma_{abs}^+$ and $\sigma_{scatt}$ which are not phenomenological. It should be remembered that sea ice, like snow, is a diffusing media and that in an optically thick sea ice the measurement of either up or downwelling light for e-folding depth is not important as has been shown by France et al. 2012 for optically thick media. It should also be noted that the limiting case mentioned by the referee for the experimentally measured upwelling radiance in this study would not be zero as the bottom of the tank is reflective and the radiative transfer modelling included the measured reflective bottom. It has been shown in France et al. 2012 that the angle at which measurements of the e-folding depth are made in an optically diffuse media has no effect on the resultant values.

**5. Confusion between radiance and irradiance? I thought that the ratio of up-welling nadir radiance to downwelling nadir radiance was being measured, but on page 12 line 13 it sounds like the measurement is ratio of upwelling irradiance to downwelling irradiance. Is there confusion here between "radiance" and "irradiance"? They are radiometrically distinct quantities and should not be interchanged. Additionally, p. 20 line 30 states that the reflectance of the laboratory grown (not 'gown') ice is considerably larger than first year ice, and resembles a reflectance closer to multi-year ice. Does this statement mean that the spectral ratio of upwelling to downwelling radiance (as far as I can tell, the only optical property measured above the ice) is being compared to spectral albedos published in the literature? Here again, I think it is possible there may be some confusion between radiance and irradiance? Or is the model being used to estimate the albedo of the ice which is then being compared to the albedos of natural ice? What natural ice measurements are being used in this comparison? Also, it would be helpful to present the measured spectral reflectivity of the tank, since it possibly matters so much.**

The comparison to literature albedos have been removed as the measurements of the albedo reported in literature would have been performed under clear skies and therefore the referee is right to point out these are incomparable to the measurements taken of the laboratory grown sea ice under diffuse conditions. The manuscript has been checked that the terms radiance and irradiance have been used correctly throughout.

The spectral nadir reflectance of the tank bottom has been added to the supplementary information and a reference to this added in the text (added here as figure 1).

**6. Figure 7 shows wavelength-dependent absorption cross-section derived from reflectance and e-folding depth data from the four runs, with no BC. I am concerned about the interpretation of these data. These curves don't really look like chlorophyll absorption spectra to me. Chlorophyll typically has absorption maxima at 430–450 nm and 640–670 nm. How was the absorption of water and ice**

**represented in the model? Is it possible there was some error in representing the ice and/or the brine, and these spectra, which look similar in nature to the absorption of water?**

The absorption spectra of chlorophyll from Bricaud et al. (2004) and chlorophyll in ice from (Mundy et al. 2011) have been added to figure 7 (now figure 9) to demonstrate the algae absorption (added here as figure 2). In previous experiments, not reported here, we have observed algae at visible concentrations in the tank.

**Minor points:**

**P. 1 L 14: As pointed out by a different reviewer in a short comment, the last line of the abstract states that "albedo is reduced by" as much as 97%. This cannot be accurate!**

The wording has been corrected to read "to" instead of "by"

**p. 1 L 22: What is the "TUV-snow radiative transfer model for sea ice"? I am not familiar with it, and I find it rather confusing that it is a "snow" model for "sea ice". What does TUV stand for?**

Further explanation of the TUV-snow model has been including here for clarity: *"The following study presents the first data from the RHUL sea ice simulator used to validate the Tropospheric ultraviolet and visible (TUV)-snow radiative-transfer model for use with sea ice. The TUV-snow model is a coupled atmosphere-snow radiative-transfer model, described in detail by Lee-Taylor and Madronich (2002). The model has been used multiple times for investigations of radiative-transfer in snow (e.g. King et al. (2005), France et al. (2011), France et al. (2012), Reay et al. (2012)) and has also been adapted for use with sea ice (e.g. King et al. (2005), Marks and King (2013), Marks and King (2014) and Lamare et al. (2016)). The model has previously been experimentally validated for photochemistry in snow by Phillips and Simpson (2005) but it has not been experimentally validated for sea ice."*

The reference to Lee-Taylor and Madronich (2002) is a full description of the model. TUV stands for tropospheric ultraviolet and uses the DISORT code of Stamnes et al. (1988). These are standard, freely available tools for radiative-transfer studies.

**p.2 L2: Here the authors mention that the sea ice simulator has "not been experimentally validated for sea ice." If that is an objective of this manuscript then it should appear perhaps in the title, and probably in the abstract. The paper is a bit diffuse because it seems to have many different objectives, as listed at the bottom of p. 2.**

The title of the paper has now been changed to reflect the main objective of the paper and the list of objectives has been removed from the paper and instead the section restructured to portray the main aims of the paper.

**p.2 L 14-15: It is not clear why validating the TUV-snow model just for a single type of sea ice, grown under particular circumstances, and a single absorber, in this case BC, necessarily means the model can be used "confidently" for other sea ice types and absorbers. For instance, I can imagine that ice grown with very few scatterers could have much smaller optical depth, and perhaps would be a different modelling problem than the one examined here.**

The suggestion the model could be used "confidently" has been toned down, with the text changed from *"If the radiative-transfer modelling with TUV-snow can reproduce the laboratory grown ice with absorbing impurities it will allow the model to be used confidently for other sea ice types and absorbers."* to read *"The work presented here will demonstrate that radiative-transfer modelling with TUV- snow (Lee-Taylor and Madronich, 2002) model can reproduce laboratory grown ices with differing fabrics with a range of mass ratios of light absorbing impurities. Such a validation will give confidence to others in the calculations of TUV-snow for other sea ices and other light absorbing impurities."*

The TUV-snow model has been applied to ablating sea ice (King et al. 2005) and to

the sea ice described in Grenfell and Maykut (1977) in Marks et al. (2013) and so has been used for other sea ice types. To the authors knowledge very few other radiative-transfer models of sea ice have been validated in a laboratory experiment, for doping with light absorbing impurities and whilst such experiments cannot cover all scenarios it gives more confidence than in an unvalidated model. If all scenarios could be validated by experiments then a model would not be required. It is the author's assertion that a model that has been successfully validated with realistic laboratory experiments is more useful than a model that has not.

**p.3 L 9: "temperature" is "higher/lower", not "warmer/cooler"**

The sentence referring to "warmer Polar temperatures" has been rewritten based on different comments from the other referee. The text now reads *"The sea ice simulator is designed to replicate a Polar sea ice growth environment under UV and visible wavelengths of solar radiation."*

**Figure 1: I see the tank volume is 2000 litres, but there is no indication of the diameter and depth? They matter, particularly in regards to the exchange of salt between the growing ice sheet and the "ocean".**

The diameter and depth of the tank were originally stated in section 2.1. For clarity they have also been added to the caption of figure 1.

**p.4 L7: does the pump achieve vertical mixing? Are you only worried about temperature stratification? What about salinity stratification?**

Yes, the pump achieves vertical mixing and therefore both temperature and salinity will not be stratified, this was the purpose of the pump. The text has now been altered to read: *"To create circulation within the tank, ensuring temperature and salinity stratification does not occur, an Iwaki…."*

**p.4 L14: what does "majority of shortwave solar wavelengths" mean? Please clarify.**

For clarity this sentence has been restructured to read *"Illumination replicating short-wave solar wavelengths over 350–650 nm is provided with a set of . . ."*

**p. 4 L18: Was the incident light field isotropic? Or just diffuse? It is difficult to create an isotropic light field in the laboratory, but it is also difficult to simulate a diffuse light field that is not isotropic in a model. Just saying they are both diffuse, does not ensure a valid comparison.**

The following text has been added to the manuscript *"The radiance, as a function of azimuth and zenith angle within the experiment was checked with a fibre optic probe and a broadband visible wavelength measurement and found to vary by 5–10%".*

**p.5 L7: Drilling a hole breaks the horizontal homogeneity of the ice block, could cause additional brine drainage, and does reduce the integrity of the ice, but the authors should be wary of stating that it "destroys the fabric of the ice", as I don't think this is accurate.**

The text has been revised from *"destroys the fabric of the ice"* to *"destroys the homogeneity of the ice"*

**p.6 L5: what size is the reflector panel? At some size it will reflect significant radiation back to the "sky" (lighting panels and white boards) and enhance the downwelling radiation field, biasing the reflectance. Please state the size of the panel and discuss the possibility of it affecting the measurement of the incident light.**

The following text has been added to the paper: *"During a reflectivity measurement a 30 cm × 30 cm Spectralon panel is added to the diffuse lighting environment above the sea ice. The addition of this panel increases the radiance, L, within the diffuse lighting environment. A very conservative estimate of the effect on the measurement of the reflectivity can be calculated by analogy to an integrating sphere. The Spectralon panel represents 0.66% of the area of the diffuse lighting environment, which is approximately*
*a cube made up of white panels and sea ice (i.e. $6 \times 1.5$ m $\times 1.5$ m $= 13.5$ m$^2$). Treating the diffuse lighting environment above the sea ice as a crude integrating sphere and considering fractional change in radiance, $\frac{\delta L}{L}$, after Ball et al 2013 who suggest $\frac{\delta L}{L} \approx \frac{A_{panel}}{A_{environment}\rho}$. Where $A_{panel}$ is the area of the Spectralon panel, $A_{environment}$ is the area of the diffusing "cube" and $\rho$ is the overall reflectivity of the diffusing cube. A very crude analysis assumes reflectivity of the panel is 1 and the part fraction of the hypothetical integrating sphere is 0. In the limit of a reflective environment $\frac{\delta L}{L} \rightarrow \frac{A_{panel}}{A_{environment}\rho} \sim 0.67\%$. Thus the overestimation of the radiance ($\sim0.67\%$) is significantly less than the uncertainty displayed on the measurement of nadir reflectivity displayed in figure 3 and figure 7."*

**p.6 L22: what does it mean that the "e-folding depth . . . is asymptotic"? please clarify. Also, it is not accurate to say "there are no near surface effects". The fact that this is a finite domain means there necessarily will be some surface effects.**

The following text has been revised for clarity *"The extinction coefficient and e-folding depth measured in the work presented here is asymptotic (reaches a constant value as shown in King et al. (2005))"*

The asymptotic region is the region where the logarithm of the radiance decreases linearly with depth and the value of the e-folding depth reaches an asymptote or constant value as shown in King et al. (2005), figure 10.

**p.7 L17: "proportion"? I think "portion" is intended?**

"Proportion" has been changed to "portion"

**p.7 L27: why two (very different!) values for the mass absorption cross-section?**

These values for the mass-absorption cross-section were obtained using different comparison materials; acetate and polypropylene sheets. The 0.58 m$^2$ g$^{-1}$ value (obtained from using the acetate sheets) has been removed from the text and was an erroneous value left in from an earlier edit for a failed determination. The acetate sheets were

unsuitable for these type of measurements.

**Table 1: units for density are not g cm$^3$. Also, I am confused about the cross-section units of cm$^2$ kg$^{-1}$. Cross-sections on previous page are cited in m$^2$ g$^{-1}$. Those are not equivalent.**

Units for density have been changed from g cm$^3$ to g cm$^{-3}$. We are keeping nomenclature of Lee-Taylor and Madronich (2002) and the body of our work (e.g. King et al. (2005), France et al. (2011), France et al. (2012), Reay et al. (2012), Marks and King (2013) and Marks and King (2014) in similar units. The units of m2 kg-1 are used for scattering cross-sections, while cm$^2$ g$^{-1}$ are used for absorption cross-sections, the authors are aware these units are not equivalent.

**Fig. 2 Y-axes have different labels should they not both be "Relative spectral absorbance"? I understand the two figures are for different materials, but I think they are intended to be compared, and if that is so then they should have the same label on their y-axes.**

The y-axis on both of these figures has been changed to "Relative spectral absorbance"

**p. 9 L 8 and following. This sentence is cryptic. It needs to be rewritten for clarity.**

The sentence has been changed to read *"Grenfell et al. (2011) showed that for small amounts of black carbon the mass loading is directly proportional to the absorbance measured by the integrating sandwich spectrometer."*

**p. 12 L 5: Why is ice density measured and reported? Is it used in the modeling? If so, the way that is used could be important and should be described.**

When describing sea ice it is normal to record its mass density (Eicken, 2003) as it can be used to calculate other properties of the sea ice as described by Weeks (2010)

**p. 12 L 10 -11: Sentence beginning "The reflectance under. . ." needs to be**

**rewritten for clarity.**

Agreed, this sentence was poorly written. The sentence has been changed from *"The reflectance under the ice is the measured, wavelength dependent, nadir reflectance of the bottom of the water filled tank"* to *"The wavelength dependant, nadir reflectance of the water filled tank is measured and included in the model as the under ice reflectance."*

**p. 13 L 4- 5: Higher air temperature should result in slower ice growth. Slower ice growth would be expected to result in less entrainment of brine within the ice. Less brine would be expected to yield fewer and/or smaller brine inclusions, which would then result in reduced scattering.**

The authors agree with the referee's logic that a larger value for the temperature of the air would result in slower ice growth, however as stated in the paper the air temperature is kept constant for all experiments.

**p. 14 L 3 -4: see comments above about reduced salinity. The data displayed in figures 5 and 8 really should be presented on the same plot; it is very difficult to make the comparison when they are in different figures.**

Figures 5 and 8 should appear on the same page of the final manuscript for easy comparison of the figures. The figures were kept separate for clarity as the figure becomes cluttered to read when plotted on one graph. Any adjustment can be made at the proof stage.

**p. 18 L 13: No, sea ice is not at its eutectic point, unless it is very cold (about –37°C). When in thermal equilibrium, it is always at its melting point, hence the required equilibrium concentrations of brine and ice.**

The reference to sea ice always being at its eutectic point has been removed from the text.

**Figure 9: please specify which y-axis corresponds with which curves.**

The figure has been changed to make it more obvious which axis corresponds to which curve (attached as figure 3 here).

**p. 19 L 4 -7 : This relates to a commonly recognised "similarity principle" in radiative transfer.**

Reference to the "similarity principle" has been added to the text.

**p. 20 L 15, following: does the exchange of seawater in the "ocean" of the simulator correct for salinity variations? I would expect even a 30 cm thick ice cover could affect the salinity of the ocean, but since the dimensions of the tank are not given (other than total volume), it is impossible to estimate the salinity enhancement in the 'ocean' due to freezing of the ice and resulting salt rejection.**

The sea ice thickness is kept thin so that the salinity of the water below the sea ice does not get too large to be unrealistic. The salinity of the underlying liquid is measured during ice growth and does increase slightly due to brine expulsion into the water.

**p. 21 L 11: Here again, ". . .reduce the albedo of the ice by 97%..." I think this should be ". . .reduce. . . to 97%".**

Correct, the text in the manuscript has been changed from "by" to "to".

**p. 21 L 15: "The derived scattering cross-section values are typical of sea ice. . ." what are the derived values being compared to?**

The following has been added to the manuscript for comparison *"The derived values of the scattering cross-section are typical of sea ice (e.g. Grenfell and Maykut 1977, Timco and Frederking, 1996, Perovich 1996), while the derived..."*

**Fig. 1.** Spectral nadir reflectance of tank bottom added to the supplementary information

[Figure]

**Fig. 2.** Updated figure for absorption cross-section of ice with absorbing impurities (bottom) showing absorption by chlorophyll (bottom)

[Figure]

**Fig. 3.** Updated figure 9 with clearer axis labels

---

## Author Comment (AC3) · 25 Aug 2017

The authors would like to thank S. Doherty for their comment on the paper "Optical properties of laboratory grown sea ice doped with light absorbing impurities (black carbon)."

**A (likely) correction to the Abstract: "Particulate black carbon at mass ratios of 75, 150 and 300 ng/g in a 5 cm ice layer lowers the albedo by 97%, 90%, and 79% compared to clean ice at a wavelength of 500 nm." I believe that the authors mean that it "lowers the albedo to (i.e. not "by") 97%, 90% and 79%" – or some other wording adjustment is needed. Lowering it *by* 97% would make for some**

**very black ice indeed!**

They are correct in their comment on the wording and this has been changed in the revised manuscript to say "to" instead of "by".

---

## Author Response (AR2)

**Review of revised manuscript entitled "Optical properties of sea ice doped with black carbon—An experimental and radiative-transfer modelling comparison"**

The authors are commended for this much improved, revised manuscript. My initial comments have mostly been satisfactorily addressed. I do however have a few comments on this revised version.

We are pleased that Referee 1 is completely happy with our paper and that Referee 2 only has technical comments that we have addressed below.

Abstract, first sentence. This run-on sentence makes it daunting for potential readers to dive into this manuscript. This first sentence needs to be rewritten.

We have broken the sentence into two smaller sentences

It was:
"Radiative-transfer calculations of the light reflectivity and extinction coefficient in laboratory generated sea ice doped with and without black carbon demonstrate that the radiative transfer model TUV-snow can be used to predict the light reflectance and extinction coefficient of sea ice typical of first year sea ice containing typical amounts of black carbon and other light absorbing impurities."

And is now:
"Radiative-transfer calculations of the light reflectivity and extinction coefficient in laboratory generated sea ice doped with and without black carbon demonstrate that the radiative transfer model TUV-snow can be used to predict the light reflectance and extinction coefficient as a function of wavelength. The sea ice is representative of first year sea ice containing typical amounts of black carbon and other light absorbing impurities."

This may be a stylistic difference, but I am not accustomed to the use of "sea ices". This terminology is used throughout the manuscript. It would seem more natural to me to see the media described as "sea ice samples".

There is nothing grammatically wrong with "sea ices" and we have left it as is since it is just stylistic and does not change the comprehension of the science in the paper. "Sea ice sample" would imply we have taken a small part of our ice, i.e sampled it and this would be a poorer description.

P 5. In the description of the illumination, it needs to be made clear whether the lights are only turned on for optical measurements or whether they remain turned on during the entire ice growth period.

The text has been changed to "Illumination, during optical measurements only, and replicating shortwave solar wavelengths over 350–650 nm is provided with a set of twenty Daystar daylight simulation fluorescent tubes and five sun-bed ultraviolet tube lights (peak illumination wavelength of ~350 nm, 40 nm FWHM).

P 15 line 9. It is not clear to me how the heat flux output from the refrigerator plant can be reduced and still maintain constant temperature in the context of this laboratory environment. If the refrigeration system is used to conduct heat out of the water in the tank, then I would expect a reduced heat flux to produce a higher air temperature in the air space above and around the tank. What am I missing?

We assume the reviewer has assumed the temperature within the simulator is determined by an equilibrium between a constant (cooling) heat flux out of the simulator from the refrigeration plant and a constant (warming) heat flux through the walls of the simulator owing to imperfect thermal insulation. A change in either of these fluxes would change the equilibrium temperature within the simulator. However such an assumption would not explain what is written in the paper. Instead like all refrigeration equipment it is normal to run it at a temperature where the temperature in the simulator was maintained by switching the refrigeration equipment on or off (or reduced and increased power). As the experiment progressed the cooling heat flux provided by the refrigeration equipment reduced requiring the refrigeration plant to remain on for longer. Thus at no time was the temperature control of the simulator changed but the heat flux from the

simulator was reduced, i.e. less heat was removed per unit area, per unit time and this compensated by increasing the time of the flux.

The text has been changed from:
"The time taken for reflectance of the ice to become a constant value became longer, taking 5 days for run 1, 7 days for run 2, 11 days for run 3 and 13 days for run 4 owing to an issue with the maintenance of the refrigerator plant which reduced its heat flux but not its maintained temperature"

To:
"The time taken for reflectance of the ice to become a constant value became longer, taking 5 days for run 1, 7 days for run 2, 11 days for run 3 and 13 days for run 4 owing to an issue with the maintenance of the refrigerator plant which reduced its heat flux from the simulator but not its maintained temperature "

Figure 9. I still don't agree with the conclusion drawn from this figure. First of all, the curve reproduced here from Bricaud (2004) doesn't show much fidelity to either Fig 1 or Fig A1a in Bricaud's original paper. Bricaud generally shows peak absorption by chlorophyll at substantially longer wavelengths (centered around 450 nm). I just don't see a basis for claiming that the residual absorption in this experiment is related to chlorophyll. This is especially true given the efforts carried out to inhibit algae growth, and the lack of natural lighting. Frankly, the absorption spectra shown in Fig. 9 b looks more like a mineral dust to me. I suppose this is largely up to our respective interpretation, and I suppose we could continue to argue the comparison, but I would suggest the authors show the residual absorption and present some various options for how it should be attributed.

We have added another panel to Figure 9 with some typical dust absorptions in ice from our paper in (Fig2b M. L. Lamare, J. Lee-Taylor, and M. D. King, Atmos. Chem. Phys., 16, 843–860, 2016 ). We believe this demonstrates that the residual absorption is not dust. The purpose of the improved figure 9 is to demonstrate our residual absorption and how we believe this could be consistent with algae - we have presented our data and published data for chlorophyll in algae and algae in sea ice. The reader can come to his or her own decision if the absorption represents absorption by chlorophyll.

Algae is extremely hard to remove completely – conditions would need to be microbiologically sterile - and this is extremely difficult to achieve in a cold room.

The reviewer correctly identified that Fig 1 of Bricaud (2004) has many absorptions of other chromophores closer to 450nm. However we have plotted the most abundant absorber from Bricaud (2004) figure 1 as detailed by Bricaud (2004). Whilst the other chromophores may be present they will be dominated by the most abundant absorber (weighted by their specific absorption cross-section) However, in section 3.2 of the same paper they highlight that with the exception of the carotenoids the amount of Chlorophyll A (and divinyl chlorophyll A), to other chromophores studied, is dominant.
It should also be noted that the chromophores of Bricaud are in organic solvent and not in algae within the ice. Such changes will shift the location of the peaks, as demonstrated in figure 9 with the plot of algae in ice from Mundy (2011)

Various axis labels show the name of the variable and then a "/" before the units. This of course looks like a "divide" sign, suggesting the units are implicitly (-1). I would recommend instead the units be placed within square brackets [ ], as I believe is the more conventional style.

The "/" is a "divide", it is not possible to plot "units" thus the quantity has to be divided through by the units before plotting the value. This notation is usually clearer than the "[]" notation as the following example will demonstrate.

Imagine the x-axis label "Ozone absorption Cross-section [$10^{20}$ $cm^2$ $molcule^{-1}$]". Has the quantity plotted been multiplied by $10^{20}$, divided by $10^{20}$ and are the units $cm^{-2}$ $moleulce^1$ or $cm^2$ $molecule^{-1}$?

Whereas "Ozone absorption Cross-section / $10^{20}$ $cm^2$ $molcule^{-1}$" is clearly a quotient and what is plotted is the ozone absorption cross-section after the measured value was divided by the quantity $10^{20}$ and the unit $cm^2$ $molecule^{-1}$.

[revised manuscript text omitted]